# Dynamic causal brain circuits during working memory and their functional controllability

Weidong Cai [1,2 ✉], Srikanth Ryali [1], Ramkrishna Pasumarthy[3], Viswanath Talasila[4] & Vinod Menon [1,2,5 ✉]

Control processes associated with working memory play a central role in human cognition, but their underlying dynamic brain circuit mechanisms are poorly understood. Here we use system identification, network science, stability analysis, and control theory to probe functional circuit dynamics during working memory task performance. Our results show that dynamic signaling between distributed brain areas encompassing the salience (SN), fronto-parietal (FPN), and default mode networks can distinguish between working memory load and predict performance. Network analysis of directed causal influences suggests the anterior insula node of the SN and dorsolateral prefrontal cortex node of the FPN are causal outflow and inflow hubs, respectively. Network controllability decreases with working memory load and SN nodes show the highest functional controllability. Our findings reveal dissociable roles of the SN and FPN in systems control and provide novel insights into dynamic circuit mechanisms by which cognitive control circuits operate asymmetrically during cognition.

[1] Department of Psychiatry and Behavioral Sciences, Stanford University School of Medicine, Stanford, CA, USA. [2] Wu Tsai Neurosciences Institute, Stanford University School of Medicine, Stanford, CA, USA. [3] Department of Electrical Engineering, Robert Bosch Center of Data Sciences and Artificial Intelligence, Indian Institute of Technology Madras, Chennai, India. [4] Department of Electronics and Telecommunication Engineering, Center for Imaging Technologies, M. S. Ramaiah Institute of Technology, Bengaluru, India. [5] Department of Neurology and Neurological Sciences, Stanford University School of Medicine, Stanford, CA, USA. ✉email: wdcai@stanford.edu; menon@stanford.edu

Almost all cognitively demanding tasks depend on working memory, the ability to maintain and manipulate information in the absence of sensory input[1–3]. Working memory is considered a "sketchpad of conscious thought" and a foundational component of information processing and problem-solving[4]. Cognitive control processes associated with working memory play an essential role in development[5,6] and impairments in these processes are a key component of cognitive dysfunction in many psychiatric disorders, including schizophrenia and attention deficit hyperactivity disorder[7–11]. Although considerable progress has been made in identifying brain regions involved in working-memory-related cognitive processes, the dynamic circuit mechanisms by which it operates in the human brain are poorly understood. In particular, little is known about dynamic causal and asymmetric interactions between distributed brain regions during working memory task performance and their cognitive load-dependent network properties. Furthermore, reliance on small sample sizes in previous studies has been a serious limitation, leaving unclear the reliability of reported findings.

Here we leverage data from the Human Connectome Project (HCP)[12] to investigate causal dynamic circuit mechanisms involving cognitive control systems consistently implicated in working memory. The present study focuses on three major concepts: causal circuits and hubs, functional controllability of brain circuits, and reproducibility of findings. We first apply state-space models to determine dynamic causal circuits, probe dynamic brain network properties including causal outflow and inflow hubs of information flow[13,14], and determine sources of individual differences in working-memory task performance. Second, we develop and apply a control theory framework extending previous work[15–18] to identify brain nodes that facilitate distinct brain states associated with high and low cognitive load. Third, we validate our findings using replicability and stability analysis. Our computational tools drawn from state-space modeling, network science, and control theory, along with a 'Big Data' approach allowed us to address fundamental gaps in our understanding of causal networks underlying cognitive control during working memory, their controllability, and reproducibility.

The first goal of our study is the identification of the directed neural circuits underlying cognition during working memory using fast temporal-resolution (subsecond) fMRI data and a large sample of well-characterized healthy young adults. The theoretical focus of the present work lies in a triple network model of cognitive control systems which posits a key role for the salience network (SN) in switching brain networks with the resultant engagement of the FPN and disengagement of the DMN during cognitively demanding tasks[13,19]. Analysis of causal control circuits has the potential to inform how asymmetries and hierarchies in directed influence allow individual networks or specific brain nodes to control others.

There is now extensive evidence demonstrating that the cognitive demanding *n*-back working-memory task engage or disengage core nodes of the salience (SN), frontoparietal networks (FPN), and default mode networks implicated in cognitive control[20–22]. Early research on the neural basis of cognitive control processes engaged during working memory has primarily focused on the dorsolateral prefrontal cortex in the middle frontal gyrus (MFG) based on observations of neuronal activity during online maintenance of task-relevant information[23–25]. Since then, non-human electrophysiological studies and human neuroimaging studies have consistently identified distributed frontoparietal regions involved in working memory including the anterior insula (AI), dorsomedial prefrontal cortex (DMPFC), anterior cingulate cortex, MFG, frontal eye field (FEF), and inferior parietal lobule

(IPL)[26–29]. Meta-analyses of human neuroimaging studies have also highlighted consistent activation of these brain areas[20,22,30,31]. Notably, these regions overlap prominently with the SN and FPN[19,32,33]. In contrast, the posterior cingulate cortex (PCC) and ventromedial prefrontal cortex (VMPFC) encompassing the default mode network (DMN) are deactivated during cognitively demanding tasks[21,34], but there have been suggestions that the degree of disengagement may have a direct impact on working-memory performance[35–37]. Other related research has suggested that the DMN may also encode task-relevant information during cognitively demanding tasks[38,39]. While these studies point to a highly consistent pattern of working-memory-related neural activity in the SN, FPN, and DMN, there have been no systematic investigations of their dynamic functional interactions and network properties.

Human neuroimaging studies of working memory have primarily focused on regional brain activation[20,22,30,31] and undirected functional interactions[37,40,41], and there is now growing interest in examining how cognitive functions emerge as a result of dynamic causal interactions between distributed brain regions[4]. The need for such investigations has gathered particular impetus in light of recent electrophysiological studies in monkeys and rodents demonstrating significant asymmetries in inter-regional interactions during working memory[4,42,43]. Crucially, the focus on undirected functional connectivity in previous studies[37,40,41] precludes inference of how brain networks or regions might influence and control other regions. More recently, analysis of causal interactions among lateral prefrontal cortex regions has pointed to a key role for the dorsolateral prefrontal cortex in integrating information for cognitive control operations[44,45]. However, a comprehensive understanding of dynamic functional interactions and causal hubs involving multiple large-scale brain networks is still lacking. Specifically, the relative order and potential hierarchies of causal signaling are unknown.

We investigated context-specific dynamic causal circuit mechanisms involving core nodes of the SN–FPN–DMN cognitive control system using multivariate dynamic state-space systems identification (MDSI)[46–48]. MDSI uses a state-space model[46–48] for estimating context-specific causal interactions in latent neuronal signals after taking into account inter-regional variations in hemodynamic response. A particular advantage of MDSI is that it does not require testing a large number of pre-specified models, which is especially problematic as the number of the models to be tested increases exponentially with the number of nodes[48]. This approach not only enabled us to probe large-scale causal networks associated with working memory, but also, critically, to determine how directed network properties such as causal hubs and network controllability change with working-memory load.

We probed network properties and casual hubs associated with context-specific interactions of the SN, FPN, and DMN. The past decade has seen an explosion of research on the study of static brain network properties based on its structural and intrinsic functional connections. In contrast, little is known about causal control hubs and how they change with working memory, or more generally, cognitive load. Hubs are highly connected regions important for integration of activity across brain regions[14,49,50]. The weighted directed graphs estimated by MDSI allowed us to compute hubs from both the perspective of outflow from, and inflow into, each node of the SN, FPN, and DMN. This enabled us to test hypotheses about the differential and asymmetric functional roles of the SN, FPN, and DMN nodes, and how they change with working-memory load.

The specific hypotheses we test in with respect were based on a body of prior work suggesting a key role for the AI node of the SN

in switching brain networks during tasks such as the stop signal, Flanker, and oddball tasks[13,51]. However, working-memory load in such response inhibition tasks is minimal and how the SN, FPN, and DMN interact causally during a canonical working-memory task such as the n-back task is poorly understood. Based on previous research we first test the hypothesis that causal interactions involving the SN, FPN, and DMN would be dynamically modulated by working-memory load and that directed network interaction patterns would differ between the high- and low-load working memory conditions. We predicted that the strength of dynamic causal interactions between the SN and FPN would increase with load, while dynamic causal interaction between the DMN and the two other networks would decrease with working-memory load. Next, based on previous findings suggesting a key role for the SN in switching networks[13,19,33], we hypothesized that the AI node of the SN would emerge as a causal outflow hub during working-memory load conditions with significant inflow into FPN and DMN nodes. Last, we hypothesized that the strength of dynamic causal interactions between the SN, FPN, and DMN would predict working-memory performance.

The second major focus of our study was the development and application of control theory to asymmetric causal circuits associated with cognition during working memory. Controllability, in the classical sense, measures the ability to perturb a system from a given initial state to random target states, in finite time, by means of external control inputs. Crucially, nodes with higher controllability require lower energy for perturbing a system from its current state[16,52], and controllability measures are useful for identifying driver nodes which have the potential to influence overall system dynamics[53]. Control properties of complex systems can provide insights into how they can be perturbed to achieve desired behaviors[54–56]. We examined average network controllability, a key index of control in complex systems which has been applied to many disciplines of physics, engineering, and biology[53,57,58]. Previous applications of control theory to human neuroimaging have been based on structural brain connectivity derived using diffusion tensor imaging[15]. However, researchers have drawn attention to the importance of dynamics between state variables for the study of controllability in complex systems[16]. Thus, structural topology by itself is ill-suited for assessing brain-cognitive-state dependent control in complex functional brain networks[59]. Here, we address this challenge using a dynamical state-space model which lends itself more naturally to assessments of controllability. Network controllability measures computed in this framework using MDSI allowed us to identify nodes that need the lowest energy to perturb the SN–FPN–DMN cognitive control system, and to determine how controllability changes with distinct brain states associated with high and low cognitive load. We predicted that network controllability would depend on working-memory load with lower controllability during high load reflecting less flexibility in network interactions. We further test the hypothesis that the AI node of the SN would emerge as the node with high controllability associated with its dominant pattern of casual outflow. We contrast our theoretical model and framing with an alternative hypothesis, based on DTI-based symmetric structural connectivity measures, that the DMN has the highest controllability in the human brain[15].

We investigated dynamic casual functional circuits using a large ($N = 737$) sample of participants from the HCP who performed an n-back working-memory task during fMRI scanning[12]. The n-back task consists of high (2-back) and low (0-back) working-memory load conditions presented in blocks. Specifically, participants were presented with a temporal sequence of visual stimuli and were required to respond whether the current stimulus was identical to the one they saw two time-steps back

(2-back) or whether the current stimuli matched a known target (0-back). The 2-back task condition places considerably higher load on working memory as it involves dynamic updating of the contents of working memory. Figure 1 provides an overview of our data analysis pipeline showing key steps including analysis of task performance, regional activation, MDSI of causal interactions during the high and low conditions, causal outflow hubs, functional controllability, and the relation between causal interactions and behavior.

Finally, the third major focus of our study was to leverage the large HCP sample size to determine the stability of our findings related to dynamic causal interactions during working memory. Previous studies of casual functional circuits engaged during working memory have been limited by small sample sizes, typically ranging from 20 to 40, often leading to inconsistent and contradictory findings[44,60–65]. Furthermore, such small sample sizes have precluded analysis of the stability and reliability of causal network interactions and network metrics. Stability analyses demonstrated the robustness of our findings, and our approach represents an important step in ensuring robustness and replication of the mechanisms by which control circuits operate during working memory.

## Results

**Behavior during the 2- and 0-back working-memory conditions**. Performance accuracy was significantly higher in the low-load (0-back) condition ($93 \pm 7\%$) compared to the high-load (2-back) condition ($86 \pm 8\%$) ($t_{736} = 23.3$, $p < 0.001$, Cohen's $d = 0.92$, two-side paired $t$-test). Reaction times on the 0-back condition ($738 \pm 136$ ms) were significantly shorter than those on the 2-back condition ($966 \pm 149$ ms) ($t_{736} = 51.5$, $p < 0.001$, Cohen's $d = 1.6$, two-side paired $t$-test). These results confirm that working-memory performance is less accurate and slower on the 2-back than 0-back conditions, consistent with the higher working-memory load on the 2-back task.

**Dynamic causal interaction patterns during the 2- and 0-back working-memory conditions**. We focused our analysis of dynamic causal interactions on 11 core SN, FPN, and DMN regions known to display a consistent profile of task-related activation and deactivation during the 2-back working-memory task[21] (Fig. 1A). SN nodes consisted of the left and right AI (lAI, rAI) and DMPFC; FPN consisted of left and right MFG (lMFG, rMFG), left and right FEF (lFEF, rFEF), and left and right IPL (lIPL, rIPL); and the DMN consisted of the PCC and VMPFC. SN and FPN regions of interest (ROIs) showed significantly greater activation in the 2-back, compared to the 0-back, task condition, whereas DMN ROIs showed significantly reduced activity in the 2-back condition (Fig. 1B). Note that the DMPFC node was identified based on its highest levels of activation in the dorsomedial wall. This peak is dorsal to the anterior cingulate cortex node that typically anchors the SN. Our designation of the DMPFC within the SN from a network identification point of view is based on several reasons: (i) independent components analysis of task and resting-state fMRI both identify an extended dorsomedial wall cluster that encompasses the DMPFC and dorsal anterior cingulate cortex[66,67]; (ii) the DMPFC showed significantly greater activation than the adjoining dorsal anterior cingulate cortex (6-mm radius sphere, $x = 7$, $y = 18$, $z = 33$), determined from a previous study using independent component analysis on a separate resting-state fMRI data[33], when contrasting the 2-back and 0-back task conditions ($t_{736} = 29.06$, $p < 0.001$, Cohen's $d = 1.07$, two-side paired $t$-test); (iii) other functional brain atlases have also assigned the DMPFC ROI to the SN[68]; and (iv) graph-theory analysis on directed connectivity matrices

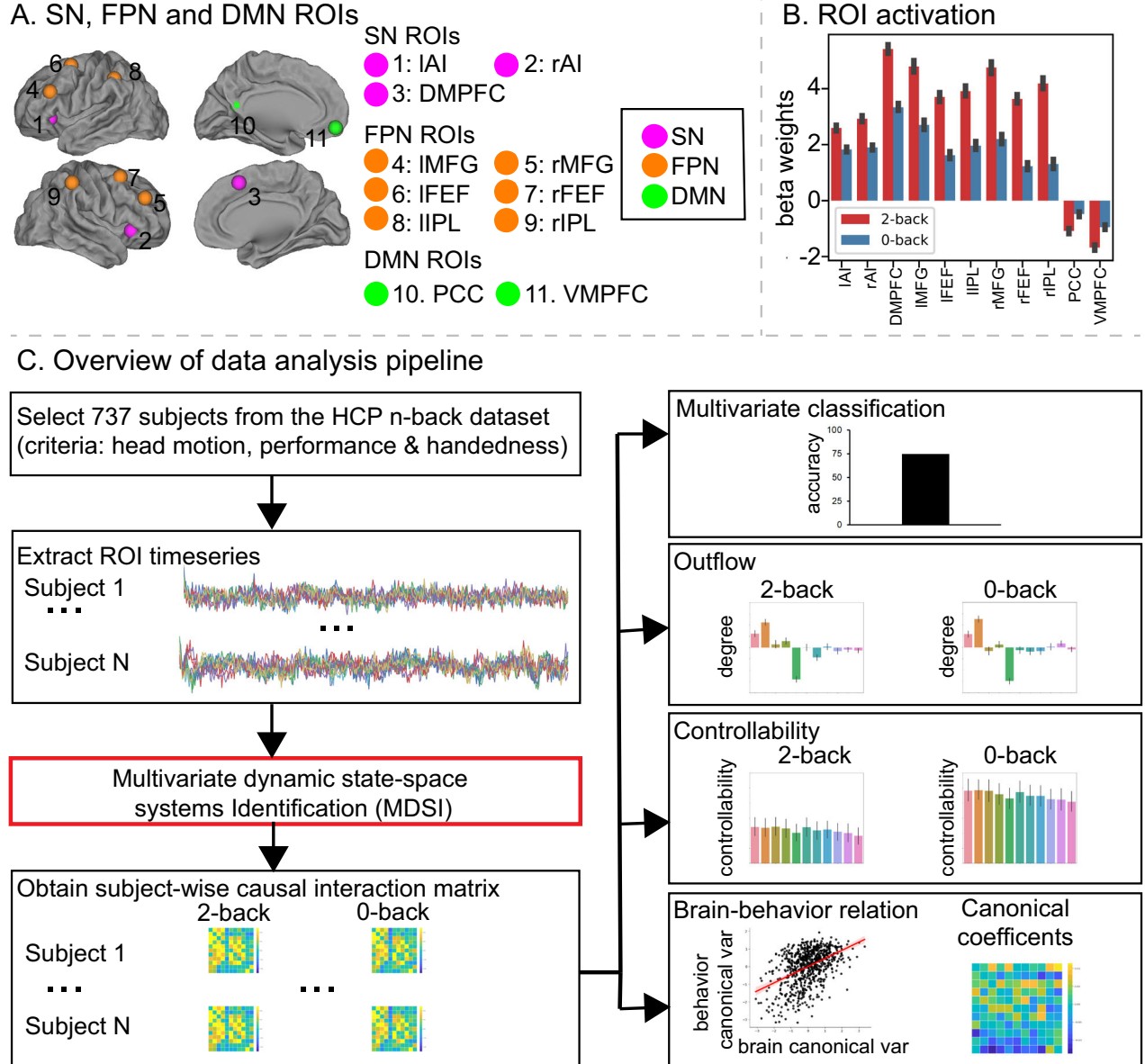

**Fig. 1 Schematic illustration of data analysis strategy and procedures. A** Salience Network (SN), Frontal-Parietal Network (FPN), and Default Mode network (DMN) ROIs: (1) left anterior insula (lAI); (2) right anterior insula (rAI); (3) dorsomedial prefrontal cortex (DMPFC); (4) left middle frontal gyrus (lMFG); (5) right middle frontal gyrus (rMFG); (6) left frontal eye field (lFEF); (7) right frontal eye field (rFEF); (8) left intraparietal lobule (lIPL); (9) right intraparietal lobule (rIPL); (10) posterior cingulate cortex (PCC), and (11) ventromedial prefrontal cortex (VMPFC). **B** General linear model analysis revealed significant working-memory-load-related activation in SN and FPN (task-positive nodes), and deactivation in the DMN (task-negative nodes) ($p <$ 0.01, FDR-corrected, two-sided $t$-test). $N = 737$ participants. Data are presented as mean ± SEM. **C** Overview of data analysis pipeline. We first screened the HCP n-back working-memory dataset based on head motion, behavioral performance, and participant handedness. We then extracted time series from each of the 11 ROIs and applied MDSI to determine working-memory load-specific dynamic causal interactions from each participant in the 2-back and 0-back task conditions. MDSI-derived causal influences were then used to investigate (1) whether multivariate dynamic causal interaction patterns distinguished 2-back versus 0-back task conditions, (2) task-dependent causal outflow from each ROI, (3) network controllability as a function of working-memory load, and (4) the relationship between the strength of dynamic causal interactions and behavioral performance. HCP human connectome project, MDSI multivariate dynamic state-space systems identification, ROI region of interest. Source data are provided as a Source data file.

estimated by MDSI demonstrated that DMPFC and AI are part of the same module during both the 2-back and 0-back task conditions (Fig. 2A).

MDSI identified several links that showed significant dynamic causal interactions in the 2-back and 0-back task conditions ($p <$ 0.01, FDR-corrected, two-side paired $t$-test) (Fig. 2B). Next, leveraging the large sample size of the HCP dataset, we examined the stability of dynamic causal interaction patterns using bootstrapping with subsamples ranging from 20 to 600. We

found that dynamic causal interaction patterns achieved a high level of stability ($r > 0.8$, Pearson's correlation) with subsample sizes of $N = 30$ or more (Fig. 2C). These results demonstrate that MDSI reliably estimates dynamic causal interaction patterns associated with both the 2-back and 0-back conditions.

**rAI is the outflow hub and rMFG is the inflow hub during both the 2-back and 0-back working-memory conditions**. We evaluated net causal influences of each node and determined causal

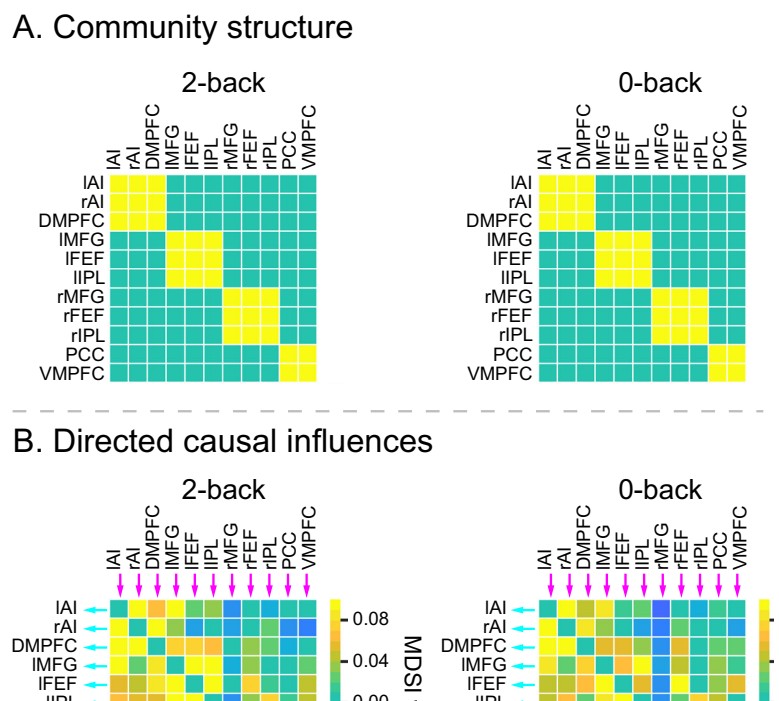

## A. Community structure

## B. Directed causal influences

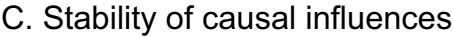

## C. Stability of causal influences

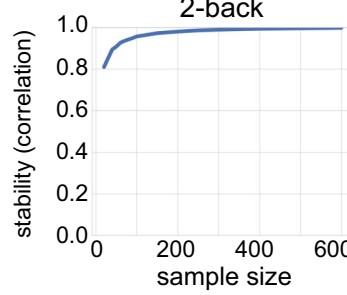
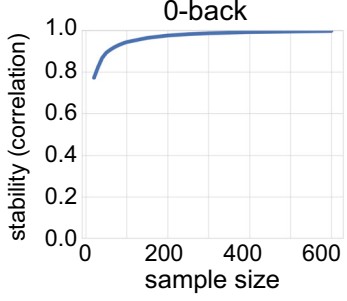

**Fig. 2 Working-memory load-specific dynamic causal influences. A** MDSI and graph-theoretic analyses identified four communities associated with directed causal influences in both the 2-back and 0-back working-memory task conditions: (i) SN consisting of lAI, rAI, and DMPFC nodes, (ii) left FPN consisting of lMFG, lFEF, and lIPL nodes, (iii) right FPN consisting of rMFG, rFEF and rIPL nodes, and (iv) DMN consisting of PCC and VMPFC nodes ($p <$ 0.01, FDR-corrected, two-sided $t$-test). $N = 737$ participants. Yellow cells indicate that a pair of ROIs are grouped into the same community and cyan cells indicate that a pair of ROIs belong to different communities. **B** Significant directed causal influences between SN, FPN, and DMN ROIs in the 2-back and 0-back working-memory task conditions ($p < 0.01$, FDR-corrected, two-sided $t$-test). $N = 737$ participants. Red cells indicate significant positive influences and blue indicates significant negative influences. **C** Stability analyses revealed highly stable multivariate patterns of causal influences among SN, FPN, and DMN nodes in 2-back and 0-back task conditions ($r > 0.8$ for sample size >25). X-axis shows sample sizes ranging from 20 to 600. Y-axis shows stability, computed as the correlation of multivariate causal influence patterns between the original sample and random subsamples drawn from $N = 737$ participants. lAI left anterior insula, rAI right anterior insula, DMPFC dorsomedial prefrontal cortex, lMFG left middle frontal gyrus, rMFG right middle frontal gyrus, lFEF left frontal eye field, rFEF right frontal eye field, lIPL left intraparietal lobule, rIPL right intraparietal lobule, PCC posterior cingulate cortex, VMPFC ventromedial prefrontal cortex. Source data are provided as a Source data file.

signaling hubs during performance of the two working-memory tasks. To accomplish this, we computed the outflow degree of each node in each task and participant. The outflow degree is the weighted node degree: averaged outflow weights (all the output connections from a node to all other nodes) minus averaged inflow weights (all the input connections to a node from all other nodes). The rAI and lAI showed significant positive outflow in both the 2-back and 0-back conditions, with the rAI showing the highest outflow degree ($p < 0.05$, FDR-corrected, Fig. 3A, two-side

paired $t$-test). Stability analysis revealed that this rAI finding was highly stable (>80%) with sample sizes of $N = 100$ or more (Fig. 3B). That is, the rAI showed the consistently highest outflow, across multiple random subsamples of the data.

In contrast, the rMFG and rFEF showed significant inflow, with the rMFG showing the highest inflow degree ($p < 0.05$, FDR-corrected, Fig. 3A, two-side paired $t$-test). Stability analysis also revealed that this rMFG finding was highly reliable (>80%) with subsample sizes of $N = 80$ or more (Fig. 3B). That is, the rMFG

## A. Directed causal outflow

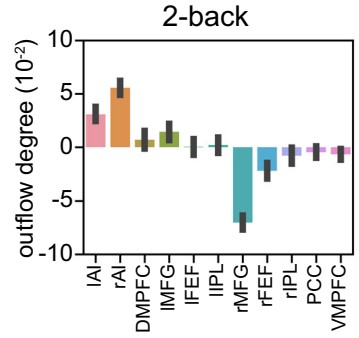
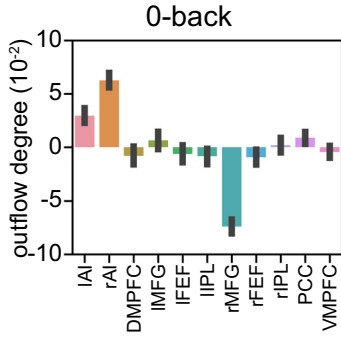

## B. Stability of causal ouflow

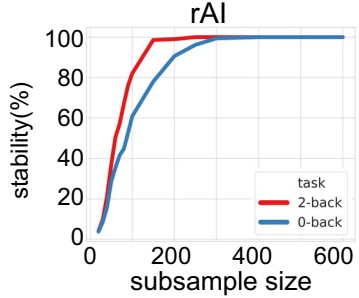
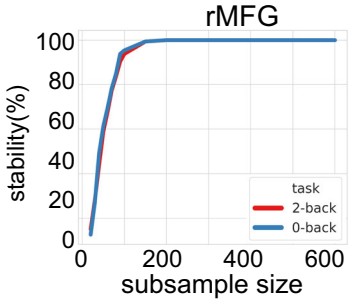

**Fig. 3 Working-memory load-specific dynamic causal outflow. A** AI showed the highest directed causal outflow between SN, FPN, and DMN nodes in both the 2-back and 0-back working-memory task conditions ($p < 0.05$, FDR-corrected, two-sided $t$-test). $n = 737$ participants. Data are presented as mean ± SEM. In contrast, the rMFG showed the highest directed causal inflow among all nodes in both task conditions ($p < 0.05$, FDR-corrected, two-sided $t$-test). $n = 737$ participants. Data are presented as mean ± SEM. **B** Stability analyses revealed highly stable directed causal outflow from the rAI and directed causal inflow into the rMFG in both the 2-back and 0-back working-memory task conditions. $X$-axis shows sample size, ranging from 20 to 600. $Y$-axis shows stability, computed as the probability that the rAI shows the highest positive directed causal outflow among SN, FPN, and DMN nodes, and the probability that the rMFG shows the highest causal inflow in random subsamples drawn from $N = 737$ participants. lAI left anterior insula, rAI right anterior insula, DMPFC dorsomedial prefrontal cortex, lMFG left middle frontal gyrus, rMFG right middle frontal gyrus, lFEF left frontal eye field, rFEF right frontal eye field, lIPL left intraparietal lobule, rIPL right intraparietal lobule, PCC posterior cingulate cortex, VMPFC ventromedial prefrontal cortex. Source data are provided as a Source data file.

showed the consistently highest inflow, across multiple random subsamples of the data.

These results identify the rAI as a robust outflow hub and the rMFG (dorsolateral prefrontal cortex) as a robust inflow hub, independent of working-memory load.

**Dynamic causal interaction patterns differentiate 2- and 0-back working-memory conditions**. We examined whether multivariate patterns of dynamic causal interactions differed between the two task conditions. A support vector machine (SVM) algorithm with 10-fold cross-validation revealed a classification accuracy of 75% ($p < 0.01$, permutation test, Fig. 4).

Next we sought to determine specific links which differ in the strength of dynamic causal interactions between the 2-back and 0-back conditions (all $ps < 0.01$, FDR-corrected, two-side paired $t$-test; Fig. 5A). Increased dynamic causal interactions in the 2-back condition were observed primarily between SN and FPN nodes. In contrast, dynamic causal influences from the PCC node in the DMN on the SN and FPN decreased in the 2-back, compared to the 0-back, condition.

We then examined the stability of working-memory load-dependent dynamic causal interaction patterns. We found that dynamic causal interaction patterns achieved a high level of

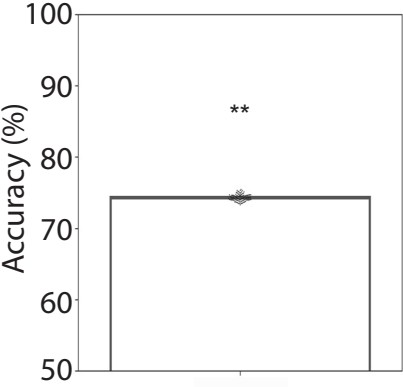

**Fig. 4 Dynamic causal influences distinguish working-memory conditions.** Linear SVM analyses with 10-fold cross-validation revealed that dynamic causal influences between SN, FPN, and DMN nodes distinguished the 2-back and 0-back working-memory task conditions ($ps < 0.01$, permutation test).

## A. Directed causal influences

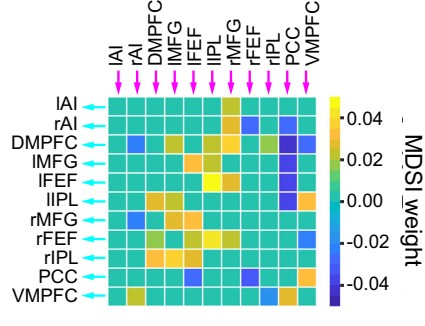

## B. Directed causal outflow

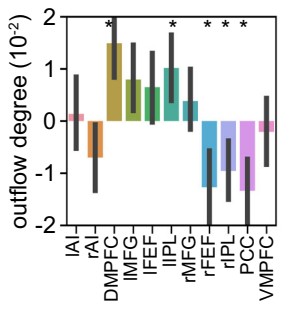

## C. Stability of causal influences

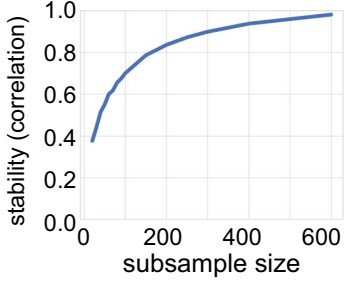

**Fig. 5 Working-memory load-dependent dynamic causal influence and net outflow. A** MDSI analysis revealed links with significantly greater directed causal influences between SN, FPN, and DMN nodes in the 2-back, compared to the 0-back, working-memory task condition ($p < 0.05$, FDR-corrected, two-sided $t$-test). $n = 737$ participants. **B** The DMPFC and lIPL showed significantly higher directed causal outflow in the 2-back, compared to the 0-back, task condition. In contrast, the PCC, rFEF and rIPL showed significantly higher directed causal inflow in the 2-back, compared to the 0-back, task condition ($p < 0.05$, FDR-corrected, two-sided $t$-test). $n = 737$ participants. Data are presented as mean ± SEM. **C** Stability analyses revealed highly stable multivariate patterns of causal influences between SN, FPN, and DMN nodes in 2-back versus 0-back ($r > 0.8$ with samples >200). $X$-axis is the subsample sizes, ranging from 20 to 600. $Y$-axis is the stability measures, which is the correlation of multivariate causal interaction patterns between subsamples and original dataset. lAI left anterior insula, rAI right anterior insula, DMPFC dorsomedial prefrontal cortex, lMFG left middle frontal gyrus, rMFG right middle frontal gyrus, lFEF left frontal eye field, rFEF right frontal eye field, lIPL left intraparietal lobule, rIPL right intraparietal lobule, PCC posterior cingulate cortex, VMPFC ventromedial prefrontal cortex. Source data are provided as a Source data file.

stability ($r > 0.8$, Pearson's correlation) with sample sizes of $N = 200$ or more (Fig. 5B). These results demonstrate that MDSI reliably estimates dynamic causal interaction patterns associated with working-memory load. Results also suggest that larger samples are needed for reliable causal interaction patterns

associated with differences between task conditions, compared to each task condition individually, and that the sample size of $N = 737$ used in the present study reliably detects modulation of dynamic causal interactions with working-memory load.

**DMPFC and PCC are dominant load-dependent outflow and inflow nodes, respectively**. We next contrasted the net causal influences of each node between the high and low-load working-memory conditions. Comparison of outflow degree between the two conditions revealed that the lAI and rAI, the two dominant outflow hubs during both the 2-back and 0-back task conditions, did not differ in net load-dependent outflow ($ps > 0.4$). Instead, it was the DMPFC, and lIPL nodes that showed significantly greater outflow in the 2-back, compared to the 0-back, task condition ($p < 0.05$, FDR-corrected, two-side paired $t$-test; Fig. 5A), with the DMPFC showing the strongest effects. Crucially, the DMPFC showed significantly greater load-dependent outflow than the lAI and rAI ($p < 0.05$, FDR-corrected, two-side paired $t$-test).

Comparison of inflow degree between the two conditions revealed that the rMFG, the dominant inflow hub during both the 2-back and 0-back task conditions, did not differ in net load-dependent inflow ($p > 0.8$, Pearson's correlation). Instead, it was the PCC, rFEF, and rIPL nodes that showed significantly higher net inflow in the 2-back, compared to the 0-back, condition ($p < 0.05$, FDR-corrected; Fig. 5B), with the PCC showing the strongest effects. Crucially, the PCC showed significantly greater load-dependent inflow than the rMFG ($p < 0.05$, FDR-corrected, two-side paired $t$-test).

These results identify the DMPFC and the PCC, rather than the AI and MFG as nodes that show load-dependent outflow and inflow during working memory.

**Dynamic causal interaction patterns predict working-memory performance**. Finally, we investigated whether dynamic causal interactions between the SN, FPN, and DMN are related to working-memory performance. A canonical correlation analysis (CCA) was used with brain measures consisting of the weights of bidirectional causal interactions and behavioral measures consisting of accuracy and reaction time in each participant. CCA model fits were significant in the 2-back condition (Pillai's trace $= 0.37$, $p < 0.05$, permutation test, Bonferroni corrected) but not in the 0-back condition (Pillai's trace $= 0.30$, $p = 0.44$, permutation test). CCA identified a significant relation between dynamic causal weights and behavioral scores in the 2-back condition ($r = 0.46$, $p < 0.001$, Fig. 6A). Figure 6B illustrates the canonical correlation coefficients, with the strongest predictive weights being those associated with the DMPFC. These results highlight the behavioral relevance of dynamic causal interactions between the SN, FPN, and DMN, and the prominent role of the DMPFC in working-memory performance.

**Network controllability differs between the 2-back and 0-back task conditions**. We used casual directed networks estimated by MDSI to investigate functional network controllability associated with the two working-memory conditions (see Supplementary Methods and Supplementary Figs. S1–S5 for detailed mathematical formulation of network controllability, and elaboration using simulated directed causal networks). Network controllability was evaluated for each node and task condition and entered into an ANOVA with factors working-memory load and node. We found a significant main effect of node ($F_{10,736} = 60.26$, $p < 2.0e{-}16$, ANOVA) and load ($F_{1,736} = 63.79$, $p < 5.33e{-}15$, AVNOA) (Fig. 7A). Network controllability was lower in the 2-back, compared to the 0-back condition, and this finding held for all nodes (all $ps < 0.001$, two-side paired $t$-test). In the 0-back task,

## A. Canonical correlation    B. Canonical coefficients

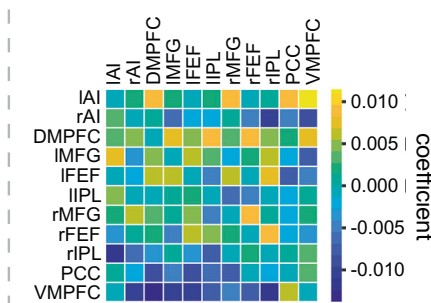

**Fig. 6 Dynamic causal influence relation to behavioral performance. A** Canonical correlation analysis revealed a significant relationship between directed SN, FPN, and DMN causal influences and behavioral performance in the 2-back working-memory task condition ($r = 0.46$, $p < 0.001$, Pearson's correlation). **B** Correlation coefficients contributing to brain-behavior relations highlights positive influences between SN and FPN nodes and negative influences of SN and FPN nodes on PCC and VMPFC nodes of the DMN. lAI left anterior insula, rAI right anterior insula, DMPFC dorsomedial prefrontal cortex, lMFG left middle frontal gyrus, rMFG right middle frontal gyrus, lFEF left frontal eye field, rFEF right frontal eye field, lIPL left intraparietal lobule, rIPL right intraparietal lobule, PCC posterior cingulate cortex, VMPFC ventromedial prefrontal cortex. Source data are provided as a Source data file.

the rAI, lAI, and DMPFC had significantly higher controllability than other nodes (all $ps < 0.001$, Bonferroni corrected, two-side paired $t$-test) except for the lFEF. In the 2-back task, the rAI, lAI, and DMPFC had significantly higher controllability than other nodes (all $ps < 0.001$, Bonferroni corrected, two-side paired $t$-test) except for the lMFG and lFEF.

To further evaluate the differential controllability of the three brain networks, we grouped ROIs' controllability scores by their networks and conducted an ANOVA with factors working-memory load and network (SN, FPN, and DMN). We found a significant main effect of network ($F_{2,736} = 60.26$, $p < 2.0e{-}16$, ANOVA), and load ($F_{1,736} = 63.79$, $p < 5.33e{-}15$, ANOVA), and a significant interaction between load and network ($F_{2,736} = 12.41$, $p < 4.53e{-}06$, AVNOA). SN nodes (rAI, lAI, and DMPFC) had the highest level of controllability in both the 0-back and 2-back tasks (all $ps < 1.9e{-}13$, two-side paired $t$-test). Network controllability was higher in the 0-back, compared to the 2-back condition, and this finding held for all three networks (all $ps < 0.001$, Bonferroni corrected, two-side paired $t$-test) (Fig. 7B). Further analysis revealed that the load × network interaction arose from load differences characterized by higher controllability of the SN compared to the FPN ($t_{736} = 4.62$, $p = 4.61e{-}06$, two-side paired $t$-test) and the DMN ($t_{736} = 3.92$, $p = 9.84e{-}05$, two-side paired $t$-test), and no differences between the FPN and DMN ($t_{736} = 0.61$, $p = 0.54$, two-side paired $t$-test).

We then examined the stability of these findings, focusing first on working-memory load-dependent differences in network controllability. Higher network controllability on the 0-back, compared to the 2-back, condition achieved a high level of stability (>80%), with sample sizes of $N = 90$ or more (Fig. 7C). Results showing network differences, with SN having the highest network controllability, also showed a high level of stability (>80%) with samples of $N = 100$ or more for the 2-back task condition and $N = 50$ or more for the 0-back condition.

These results demonstrate that network controllability decreases with working-memory load, and that SN nodes have the highest controllability among SN, FPN, and DMN nodes. Furthermore, these results are highly stable and reliable.

**Working-memory load effects on outflow and inflow.** Additional analysis examined the effects of working-memory load on outflow and inflow separately. This analysis revealed distinct patterns of outflow and inflow weights associated with memory load (Supplementary Notes A, Supplementary Fig. 6). Notably,

the SN and FPN showed significantly greater outflow in the 2-back than 0-back conditions whereas the DMN had significantly weaker outflow in the 2-back than 0-back condition ($ps < 0.05$, FDR-corrected, two-side paired $t$-test). In contrast, the FPN, but not the SN or DMN, showed significantly greater inflow weights in the 2-back than 0-back condition ($ps < 0.05$, FDR-corrected, two-side paired $t$-test).

**Reproducibility of findings in subsamples.** To examine the robustness of our findings with respect to subsamples, we divided the data into three subsets and conducted a complete set of supplemental analyses on each subset. We replicated all our findings as described in Supplementary Information (Supplementary Notes B, Supplementary Figs. 7–9).

**Hierarchy of network controllability with respect to left and right FPN.** Additional analyses revealed a hierarchy of network controllability such that controllability was significantly greater in SN than LFPN, in LFPN than RFPN, and in RFPN than DMN (all $ps < 0.01$, Bonferroni corrected, two-side paired $t$-test).

**Robustness of network controllability with respect to normalized causal interaction weights.** MDSI simultaneously estimates the causality between regions (**C** matrices) under each condition within the same modeling framework. Therefore, the average controllability estimated under the different conditions can be directly compared without the need for additional normalization. To further ensure that our findings were robust with respect to mean connection strength in each condition, we conducted network controllability analyses on normalized causal interaction weights. We replicated all our findings (Supplementary Notes C, Supplementary Fig. 10).

**Robustness of findings with respect to ROI selection.** To examine the robustness of our findings with respect to node selection, we conducted comprehensive analyses using functional ROIs determined using meta-analysis of working-memory tasks. We replicated all our findings (Supplementary Notes D, Supplementary Figs. 11–16).

## Discussion
We used computational tools drawn from state-space modeling, network science, and control theory to investigate dynamic circuit mechanisms underlying working memory in a core cognitive

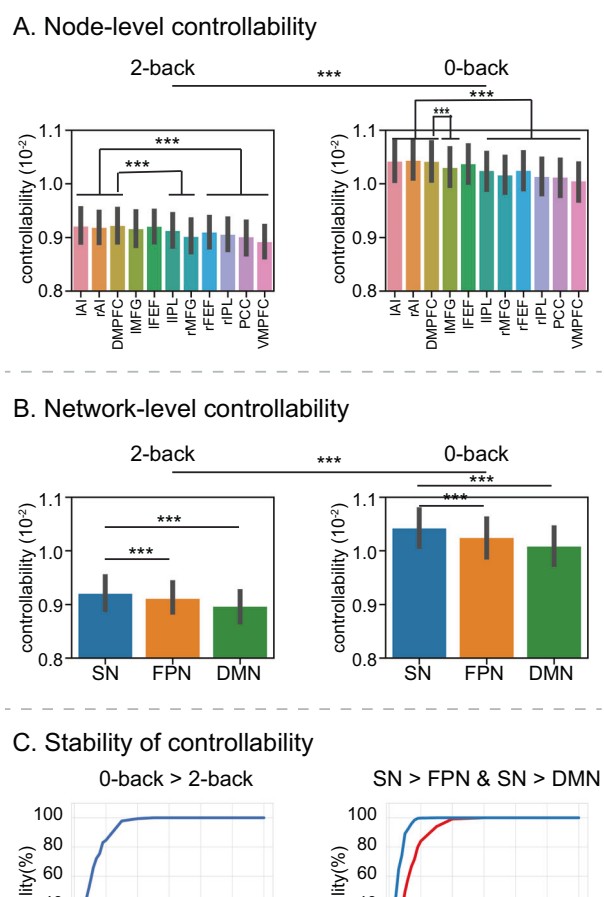

**Fig. 7 Working-memory load-dependent functional controllability in SN, FPN, and DMN. A** Functional network controllability, assessed in each brain node, was significantly lower in the 2-back, compared to the 0-back, working-memory task condition ($p < 0.001$, two-sided $t$-test). $n = 737$ participants. Data are presented as mean ± SEM. AI and DMPFC nodes in the SN have significantly higher controllability than FPN and DMN nodes, except for lFEF and lMFG nodes of the FPN in the 2-back task condition and lFEF in the 0-back task condition ($p < 0.001$, two-sided $t$-test). $n = 737$ participants. Data are presented as mean ± SEM. **B** Functional network controllability, assessed across SN, FPN, and DMN nodes, was significantly lower in the 2-back, compared to the 0-back, working-memory task condition ($p < 0.001$, two-sided $t$-test). $N = 737$ participants. Data are presented as mean ± SEM. The SN shows significantly higher controllability than the FPN ($p = 4.61e−06$, two-side paired $t$-test) and DMN ($p = 9.84e−05$, two-side paired $t$-test). $N = 737$ participants. Data are presented as mean ± SEM. **C** Stability analyses revealed stable load effect (0-back > 2-back) and network difference (SN > FPN and SN > DMN). $X$-axis shows sample size, ranging from 20 to 600. $Y$-axis shows stability, computed as the probability that the load effect of controllability is significantly different between 2-back and 0-back working-memory conditions, and the probability that the SN shows greater controllability than the FPN and DMN in both 2-back and 0-back working-memory conditions, in random subsamples drawn from $N = 737$ participants. lAI left anterior insula, rAI right anterior insula, DMPFC dorsomedial prefrontal cortex, lMFG left middle frontal gyrus, rMFG right middle frontal gyrus, lFEF left frontal eye field, rFEF right frontal eye field, lIPL left intraparietal lobule, rIPL right intraparietal lobule, PCC posterior cingulate cortex, VMPFC ventromedial prefrontal cortex. Source data are provided as a Source data file.

control system anchored in the salience, frontoparietal, and default mode networks (SN, FPN, and DMN, respectively). We uncovered robust and stable load-dependent dynamic causal interaction patterns associated with working memory. Dynamic causal influences involving the SN, FPN, and DMN accurately distinguished between the high and low-load conditions and predicted behavioral performance. Network analysis of directed causal influences revealed that the anterior insula node of the SN is a casual outflow hub whereas the dorsolateral prefrontal cortex node of the lateral FPN is an inflow hub, independent of working-memory load. In contrast, the DMPFC emerged as a dominant load-dependent outflow node with stronger load-dependent outflow than both the right and left AI. From the perspective of hierarchical signaling, our findings highlight the relative control properties of the SN and FPN and bring to light the importance of the SN in driving control.

We further developed a control-theoretic framework to investigate working-memory load-dependent functional network controllability and to identify driver nodes which influence overall system dynamics. We found that network controllability decreased with working-memory load, and that the anterior insula and DMPFC nodes of the SN showed the highest network controllability. Finally, we leveraged a large sample of participants ($N = 737$) to demonstrate the reliability of our findings. Our findings provide insights into dynamic circuit mechanisms by which core cognitive control circuits operate asymmetrically during working memory, and highlight the dissociable roles of the SN, FPN, and DMN in cognitive systems control.

**Dynamic causal interactions in the SN–FPN–DMN cognitive control system distinguish high and low-load working-memory conditions.** Dynamic causal interactions between SN, FPN, and DMN nodes reliably distinguished between the high- and low-load working-memory conditions. At the network level, classification analysis with cross-validation revealed that multivariate patterns of dynamic causal interactions accurately distinguished between and predicted high (2-back) versus low (0-back) working-memory task conditions (Fig. 4). Stability analysis revealed that this result was reliable and replicable across subsamples.

We then identified individual links that distinguish between the 2-back and 0-back conditions. We found that dynamic causal influences between multiple SN and FPN nodes that were significantly greater in the 2-back, compared to the 0-back, conditions (Fig. 5A). In contrast, dynamic causal interactions linking the DMN to the SN and FPN were lower in the 2-back, compared to the 0-back, task conditions. Thus, dynamic causal influences within the SN and FPN ("task-positive" networks) increased with working-memory load, whereas dynamic causal influences from the DMN ("task-negative" network) to the SN and FPN decreased with working-memory load. These results demonstrate how modulation of cognitive load during working memory alters causal dynamics in the SN–FPN–DMN cognitive control system.

**Causal outflow and inflow hubs during working memory.** MDSI-derived measures of the full bidirectional (asymmetric) connectivity matrix between SN, FPN, and DMN nodes allowed us to probe key network properties associated with working memory. We identified casual outflow and inflow hubs during the 2-back and 0-back task conditions by computing the net weighted degree of each node (Fig. 3A). A high positive node degree indicated that a node exerted greater causal influence on other nodes than other nodes exerted on it, while a high negative value indicated the reverse. This analysis revealed that the rAI and lAI

had the highest causal outflow during both task conditions. In contrast, the rMFG was the inflow hub, with the strongest causal influences on this node from all other regions, again, in both task conditions. Crucially, net causal influences did not differ in these regions between the high and low-load conditions, and stability analysis demonstrated the robustness and replicability of these findings.

The AI, a key node in the SN, is engaged in a wide range of cognitive control[69,70], and it has an important role in detecting salient external signal, relocating cognitive resources, and directing goal-directed behavior[19]. While the AI has not been widely probed in the context of working-memory, meta-analytic studies have shown that the AI is activated during almost all working-memory tasks[20,31,71]. Interestingly, engagement of the AI does not appear to be dependent on the category of information maintained in working memory, including social and/or affective information[72], visual objects[21], spatial location[73], and verbal and tonal stimuli[74]. Yet the role of AI in human working memory has remained elusive.

Our MDSI analysis suggests that the AI plays a dominant role in driving dynamical interactions among the SN, FPN, and DMN, independent of working-memory load. This finding is consistent with its hypothesized role in transient network switching in response to salient behaviorally relevant stimuli[13,19,33]. The 0-back task relies on detection of a stimulus that matches the target whereas the 2-back task requires participants to not only detect and encode incoming stimuli, but also to maintain and update information. The dominant causal influence of the AI on the SN, FPN, and DMN is thus more closely aligned to detection and transient encoding of stimuli, a process shared by both the 2-back and 0-back working-memory tasks. To our knowledge, there are no electrophysiological investigations of the primate AI during performance of working-memory tasks. However, a recent optogenetic study in rodents suggests a specific causal role for the AI in working memory: suppression of transient delay period activity in the AI significantly reduced working-memory task performance[75].

Our findings further highlight a dissociation between the functional roles of the AI and the MFG node of the FPN, which encompasses dorsolateral prefrontal cortex regions consistently implicated in working memory[22]. Remarkably, while the AI was an outflow hub, the MFG was reliably identified as an inflow hub during both the 2-back and 0-back task conditions. These results help resolve an important unaddressed issue regarding the differential roles of the AI and MFG in working memory. We suggest that higher causal inflow into the MFG may activate MFG regions known to play a critical role in maintenance and manipulation of the content of working memory. High inflow may reflect convergence of signals associated with external stimuli, internal representations, or task rules needed for working memory performance. Further studies examining spatiotemporal dynamics with simultaneous electrophysiological recordings are required to test this hypothesis.

**Working-memory load-dependent net outflow and inflow.** Next, we examined whether the lAI and rAI, the two dominant outflow hubs during both the 2-back and 0-back task conditions, differed in net load-dependent outflow. Notably, both the left and right AI showed no differences with working-memory load. Instead, analysis of the net weighted degree revealed that the DMPFC had the highest differential outflow associated with working-memory load (Fig. 5B). Moreover, DMPFC net outflow was significantly greater than both the left and right AI even after correction for multiple comparisons.

The DMPFC has been implicated in a broad range of cognitive control tasks[70,76,77] and damage or disturbance in the DMPFC impairs cognitive control functions[78,79]. However, the functional role of this region has remained unclear as its dynamic causal interactions have not been previously probed. The 2-back working-memory task, in particular, places significant demands on cognitive control as it requires dynamic updating of information with each new stimulus presentation. Our identification of higher outflow in the 2-back, compared to the 0-back, condition is consistent with neurophysiological studies demonstrating (i) early modulation of DMPFC activity by working-memory load[80], (ii) engagement during switching from automatic to controlled action[81], and (iii) selection of action sets[77], the demand for all of which is significantly greater in the 2-back task condition. Our findings are also aligned with a recent optogenetic investigation in rodents showing that modulation of projection from medial prefrontal cortex to the AI impairs circuit plasticity during working memory[75]. Together, these findings suggest that the DMPFC plays an important role in reconfiguration of functional circuits with increased working memory task demands.

We then examined whether the rMFG, the dominant inflow hub during both the 2-back and 0-back task conditions, differed in net load-dependent inflow. Notably, although the rMFG showed no differences in net causal flow with working-memory load, its dynamic causal interactions with multiple SN and FPN nodes increased with working-memory load (Fig. 5A). Instead, analysis of the net weighted degree revealed that the PCC had the highest differential inflow associated with working-memory load (Fig. 5B). Moreover, PCC net inflow was significantly greater than the rMFG even after correction for multiple comparisons. Thus, in contrast to the DMPFC, the PCC showed significantly increased net negative inflow in the 2-back, compared to the 0-back, task condition (Fig. 5B). This parallels the reduction of PCC activation during the 2-back, compared to the 0-back task condition (Fig. 1B).

The PCC is a core node of the DMN, which is involved in self-referential processes including mind-wandering, stimulus-independent thoughts, and autobiographic memory[82–86]. Increased cognitive demand is often accompanied by DMN deactivation across a broad range of cognitive tasks, including working memory[87,88]. Our findings demonstrate that deactivation of the PCC is accompanied by net negative causal outflow from the PCC (Supplementary Fig. 6). Reduced outflow signals and consistent with disengagement of the DMN from the SN and FPN, which facilitates access to working-memory resources required for task performance[35]. The inability to generate these causal inflow signals may contribute to working-memory deficits observed in patients with schizophrenia and depression[89,90].

**Dynamic causal interactions predict working-memory performance.** Dynamic causal mechanisms identified by our study are behaviorally relevant. We used CCA to examine the relation between the strength of dynamic causal interactions and behavioral performance in the 2-back task. We found a significant canonical correlation model and a strong relationship between multivariate measures of task-evoked causal connectivity and behavioral variables in the 2-back condition. Interestingly, the CCA model showed a significant model fit only in the 2-back condition but not in the 0-back condition, demonstrating the specificity of findings with respect to working-memory load. Predictive weights consisted of positive directed connectivity between the SN and FPN and negative directed weights between the DMN, and the two "task positive" networks (Fig. 6B). Findings thus implicate dynamic causal interactions between all three

networks in successful working-memory performance, and further demonstrate involvement of the DMN together with the SN and FPN in influencing complex cognitive functions.

**Network controllability during working-memory performance**. Analysis of the control properties of complex networks has the potential to provide insights into how they can be manipulated to achieve desired behaviors[54–56,58]. We used average controllability, measured as the trace of controllability Gramian $W_t$ as our quantitative metric for controllability[17]. In the context of functional brain networks, average controllability quantifies the influence of each node or module over the entire network, with higher controllability reflecting lower average control energy needed to drive networks from a given node or a set of nodes (see Supplementary Methods). Our approach here examines cognitive context-dependent controllability and emphasizes the importance of the actual system dynamics in determining control[91,92].

Two key findings emerged from our analysis of load-dependent functional controllability during working-memory task performance. First, controllability decreased with working-memory load (Fig. 7). This suggests that higher input energy is required to control the network when the working-memory load is high. This finding is noteworthy because it shows that network controllability is load and context-dependent; in other words, functional brain circuits are more difficult to control during complex, compared to simpler, cognitive tasks. Notably, these findings were highly reliable and remained stable with sample sizes of $N = 90$ or more. We suggest that with more items to manipulate in working memory, network interactions become more rigid thus reducing controllability. It should be noted, however, that controllability is not directly related to the overall magnitude of coupling but rather the structure of the connectivity matrix[93].

Second, we found that the AI and DMPFC nodes of the SN had the highest controllability, implying that they are candidate nodes for controlling network dynamics with the lowest levels of input energy. Moreover, these SN nodes showed the highest controllability independent of working-memory load; that is, SN nodes are adequate for controlling network dynamics under both the high and low conditions. In addition to determining controllability associated with each node, we also examined controllability associated with the SN, FPN, and DMN. Here we took advantage of the observation that any linear function of the controllability Gramian is a modular set function[18]. This implies that we can first compute the Gramian of each node individually and then average across the nodes of each network to determine controllability of each network. This analysis revealed that SN nodes had significantly higher controllability than the FPN and DMN. Additional analysis revealed that in both the 2-back and 0-back tasks, controllability was significantly greater in the SN compared to left FPN, in the left FPN compared to right FPN, and in right FPN compared to the DMN. These results point to hemispheric differences and a hierarchy of controllability in lateral frontoparietal cortex. Whether this pattern reflects a control hierarchy from attention capture (SN) to abstract control (left FPN) to concrete control (right FPN) is an intriguing hypothesis that warrants further investigation[44]. Nodes or modules with high average controllability are predicted to have a larger influence over the network. The high network controllability of the AI at the node level and the SN at the network level is consistent with their critical role in dynamic network switching[33,94] and may contribute to its high levels of spatiotemporal flexibility observed across a wide range of cognitive tasks[13,19,51,95].

Our findings demonstrate that functional network controllability is dependent on cognitive load, and furthermore, reveal that SN nodes have the highest functional network controllability. We hypothesize that these nodes are prime targets for altering functional network dynamics.

**Robust mechanisms underlying triple network model of dynamic cognitive control**. The present study has focused on a theoretical model of prefrontal cortex networks that support flexible cognitive control functions[13,19]. Our findings based on asymmetric patterns of directed causal influence between key nodes of the SN, FPN, and DMN as well as their functional controllability are consistent with the triple network model of cognitive control, and shed further light on the underlying mechanisms and their robustness. First, the AI showed the highest causal outflow among all nodes suggesting a dominant role of this key SN node in driving dynamic interactions between regions between the SN, FPN, and DMN. Second, high levels of causal inflow into the MFG suggest that this FPN node is crucial for integrating cognitive control signals from multiple other prefrontal and parietal cortical regions. Third, disengagement of the DMN from the SN and FPN under high cognitive load is marked by reduced outflow signals from the PCC. Fourth, the SN demonstrated the highest controllability, indicating a prominent role in driving network interactions. Fifth, these findings were replicable and stable, as discussed below. Taken together, these findings provide insights into the mechanisms by which prefrontal cortex networks implement cognitive control and facilitate rapid system reorganization to meet cognitive task demands. Further studies are needed to clarify the role of SN, FPN, and DMN in the context of models of hierarchical control associated with different levels of abstraction[96] and multiple-demand systems engaged by diverse cognitive demands[97].

**Reproducibility and detection of stable dynamic causal connectivity patterns during working memory**. Reproducibility is a major challenge for all neuroscience[98,99]. This is especially true for computational models that infer causal dynamics using fMRI[13,100,101]. To our knowledge, no previous studies have addressed this question in any cognitive domain using large samples to evaluate the stability of causal dynamics. Here we addressed this challenge by leveraging the large sample size of the HCP data, which allowed us to probe the stability of dynamic causal connectivity and network properties during working memory. We used bootstrapping procedures to determine the stability of findings as a function of sample size. This approach revealed replicability of our findings, and identified sample sizes required to achieve reliability: (1) multivariate patterns of dynamic causal interactions in the 2-back and 0-back conditions were highly stable with sample sizes of 30 or more; (2) stable estimates of the differences between high and low working-memory load conditions required large samples of 200 or more; (3) the rAI and rMFG emerged as robust outflow and inflow hubs, respectively, with sample sizes of $N = 80–100$ or more; (4) network controllability decreases with working-memory load and SN nodes having the highest controllability was stable with sample sizes of $N = 100$ or more. Together these findings suggest that a sample size of 737 uncovers stable patterns of dynamic causal interactions associated with the n-back task. An important implication of our findings is that sample sizes used in most previous fMRI studies of causal circuit interactions may be problematic and may have led to highly inconsistent findings reported in the literature, particularly with respect to working-memory load-dependent effects. While our findings stress the importance of establishing reliability in analysis of causal circuit dynamics, the question of sample size ultimately depends on experimental design and sufficient individual-level data[102].

**Note on interpreting causality and the effects of unobserved variables**. The present study has leveraged advances in state-space models to jointly infer causal interactions between brain regions without the confounding influences of regional variation in hemodynamic response function or the need to test multiple network models[47]. While this provides distinct advantages over techniques such as Granger causal analysis, dynamic causal modeling, and transfer entropy, two issues need to be highlighted. First, at present no (validated) methods exist that can extend causal network analysis, controlling for hemodynamic confounds, to the whole-brain level, spanning 350 or more anatomically distinct regions, due to the sheer number of parameters that need to be estimated. Instead, our approach has focused on probing causal dynamics associated with three key prefrontal cortex networks consistently implicated in cognitive control and working memory. Second, relatedly, this raises the question of whether findings could be influenced by unobserved confounds because erroneous inferences on connectivity (causal or otherwise) can occur when data from a limited set of brain regions or neuronal populations are used in data analysis[103], a problem that is further confounded when there is a mismatch between the true network dynamics and the model used for inference[104]. There currently are no good solutions to these problems, short of extensive invasive manipulations to each processing unit spanning the entire brain. We have addressed this challenge in the present study by, as summarized in the previous section, conducting extensive reproducibility, stability, and cross-validation analyses using a large sample size, the only realistic approach with real-world non-invasive human brain imaging data.

Asymmetries in directed influence are the essence of how one brain region controls another. State-space modeling and network analysis uncovered mechanisms underlying operation of a core SN–FPN–DMN cognitive control circuit implicated in working memory. Our analyses revealed that causal influences between multiple nodes in the FPN, SN, and DMN are modulated by working-memory load, and predict working-memory performance. We identified casual outflow and inflow hubs reflecting asymmetries in how core cognitive control circuits operate during working memory. Functional network measures enabled us to determine how directed network properties such as causal hubs and network controllability change with working-memory load, revealing the dissociable roles of the SN and FPN in cognitive systems control. Importantly, we demonstrate high levels of reliability of our findings using subsampling techniques and provide unique insights into reproducible dynamical systems-based mechanisms of human working memory. More broadly, our computational approach drawing on state-space modeling, network science, and control theory provide tools for probing working memory and cognitive control in the human brain, and their dysfunction in psychiatric and neurological disorders.

## Methods

**Ethics statement**. Data acquisition for the HCP was approved by the Institutional Review Board of The Washington University in St. Louis (IRB # 201204036), informed consent was obtained for each participant, and data were de-identified.

**HCP data selection**. HCP data from 737 right-handed individuals (age: 22–36 years old, 413 female/324 male) were selected from a total of 1200 subjects based on the following criteria: (1) participant had complete n-back task behavioral and fMRI data; (2) range of head motion in any translational and rotational direction <1 voxel; (3) average scan-to-scan head motion <0.2 mm; (4) accuracies in 0-back and 2-back conditions >50%; and (5) criterion (1)–(4) met in both sessions separately.

**HCP n-back working-memory task**. The HCP n-back working-memory task combines the category-specific representation task and the n-back working-memory task in a single-task paradigm[105]. Subjects were presented with blocks of trials that consisted of pictures of faces, places, tools, and body parts. Within each session, the four different stimulus types were presented in separate blocks.

Furthermore, within each session, half of the blocks are 2-back working-memory tasks and half are 0-back working-memory tasks. In the 2-back working-memory task blocks, subjects were requested to determine whether the current stimulus matches the stimulus in two presentations of stimuli prior within the same block. In the 0-back working-memory task blocks, subjects were requested to determine whether the current stimulus matches the target that was presented in the beginning of each block (cue). A 2.5 s cue indicates the task type (and target for the 0-back task) at the beginning of each block. Each of the two sessions contains 8 task blocks (10 trials of 2.5 s each, for 25 s) and 4 fixation ("rest") blocks (15 s). On each trial, the stimulus is presented for 2 s, followed by a 0.5-s inter-trial-interval (ITI).

**fMRI data acquisition**. For each individual, 405 frames were acquired in each session using multiband, gradient-echo planar imaging with the following parameters: TR = 720 ms; TE = 33.1 ms; flip angle = 52°; field of view = 280 × 180 mm; matrix = 140 × 90; and voxel dimensions = 2 mm isotropic.

**fMRI preprocessing**. Raw fMRI data for both sessions were obtained from the HCP and underwent standard preprocessing steps, including realignment, slice-time correction, normalization, and spatial smoothing with a Gaussian kernel of 6-mm FWHM[70] using SPM12 (https://www.fil.ion.ucl.ac.uk/spm/software/spm12/).

**General linear model and contrasts of interest**. A conventional general linear model (GLM) analysis was conducted in order to determine load-dependent and categorical-dependent activation/deactivation peaks. Each block in each session was modeled as one of the following vectors: 0-back-faces, 0-back-places, 0-back-tools, 0-back-body, 2-back-faces, 2-back-places, 2-back-tools, and 2-back-body. The onset and duration of each vector were the onset and duration of the corresponding block. The contrasts of interest include 0-back, 2-back, and 2-back versus 0-back.

**Networks and regions of interest (ROI)**. We used SN, FPN, and DMN nodes from a previous study based on the same HCP n-back working-memory task as used here, albeit with a smaller sample size ($N = 122$)[21]. SN and FPN ROIs were identified based on working-memory load-dependent activation (2-back > 0-back task conditions): lAI, rAI, lMFG, rMFG, lFEF, rFEF, lIPL, rIPL, and DMPFC. DMN ROIs PCC and VMPFC were based on task-related deactivation (2-back < 0-back) (Fig. 1A). Each ROI was constructed as a 6-mm radius sphere centered at the voxel that showed peak activation.

**ROI activation analysis**. Contrast weights of 2-back and 0-back task conditions were extracted in each ROI. Paired t-tests were used to examine whether activations differed significantly between the two task conditions and corrected for multiple comparisons using Bonferroni correction.

**Time series extraction**. The original time series were extracted from the preprocessed fMRI data for each ROI, resulting in a matrix with a dimension of $T \times N$, where $T$ is the number of time points and $N$ is the number of voxels in the ROI. Singular value decomposition was applied on the ROI time series matrix, and the resultant first eigenvector corresponding to the first principal component is obtained to represent the signals of interest within the ROI. The output was a $T \times 1$ vector. We used the first eigenvariate instead of mean signal within the ROI to reduce noise in potentially heterogeneous ROIs. A multiple linear regression approach with 6 realignment parameters (3 translations and 3 rotations) was applied to the time series to reduce head-motion-related artifacts and high-pass filtered (>0.008 Hz).

**Multivariate dynamical systems identification of causal interactions**. We used multivariate dynamical systems identification (MDSI) to investigate causal interactions in the SN, FPN, and DMN during working memory. Here we provide brief descriptions of the method. Details can be found in the publications focused on method development[46–48], and scripts are available from SCSNL website (https://med.stanford.edu/scsnl/publications.html).

MDSI is a state-space model consisting of a state equation to model the latent "neuronal–like" (quasi neuronal) states of the dynamic network and an observation equation to model BOLD-fMRI signals as a linear convolution of latent neural dynamics and hemodynamic responses[48]. MDSI estimates both intrinsic and experimentally modulated causal interactions between brain regions while accounting for variations in hemodynamic responses in these regions.

The state equation in MDSI is a multivariate linear difference equation or a first-order multivariate auto regressive (MVAR) model that defines the state dynamics

$$\mathbf{s}(t) = \sum_{j=1}^{J} v_j(t) \mathbf{C}_j \mathbf{s}(t-1) + \mathbf{w}(t) \tag{1}$$

The model for the observed BOLD responses is a linear convolution model

$$\mathbf{x}_m(t) = \left[ \mathbf{s}_m(t) \mathbf{s}_m(t-1) \ldots \mathbf{s}_m(t-L+1) \right]' \tag{2}$$

$$y_m(t) = \mathbf{b}_m \boldsymbol{\Phi} \mathbf{x}_m(t) + e_m(t) \tag{3}$$

In Eq. (1), $\mathbf{s}(t)$ is a $M \times 1$ vector of latent signals at time $t$ of $M$ regions, $\mathbf{C}_j$ is task-specific $M \times M$ connection matrix and $v_j(t)$ is the $j$-th experimental condition

at time $t$. $\mathbf{C}_j(m,n)$ denotes the strength of causal connection from $n$-th region to $m$-th region for the $j$-th task. $\mathbf{w}(t)$ is an $M \times 1$ state noise vector that is assumed to be Gaussian distribution with covariance matrix $\mathbf{Q}(\mathbf{w}(t) \sim N(0, \mathbf{QI}))$, where $\mathbf{I}$ is an identity matrix of size $M \times M$. In addition, state noise vector at time instances 1, 2, ..., $T$ ($\mathbf{w}(1)$, $\mathbf{w}(2)$, ...$\mathbf{w}(T)$) are assumed to be identical and independently distributed (iid). Equation (1) represents the time evolution of latent signals in $M$ brain regions. The latent dynamics modeled in Eq. (1) gives rise to the observed fMRI time series represented by Eq. (3).

We model the fMRI time series in region "$m$" as a linear convolution of HRF and latent signal $\mathbf{s}_m(t)$ in that region. To represent this linear convolution model as an inner product of two vectors, the past $L$ values of $\mathbf{s}_m(t)$ are stored as a vector. $\mathbf{x}_m(t)$ in Eq. (2) represents an $L \times 1$ vector with $L$ past values of latent signal at $m$-th region.

In Eq. (3), $y_m(t)$ is the observed BOLD signal at time instance $t$ for $m$-th region. $\boldsymbol{\Phi}$ is a $p \times L$ matrix whose rows contain bases for hemodynamic response function (HRF). Here, we use the canonical HRF and its time derivative as bases, as is common in most fMRI studies. $\mathbf{b}_m$ is a $1 \times p$ coefficient vector representing the weights for each basis function in explaining the observed BOLD signal $y_m(t)$. Therefore, the HRF in $m$-th region is represented by the product $\mathbf{b}_m\boldsymbol{\Phi}$. The BOLD response in this region is obtained by convolving HRF ($\mathbf{b}_m\boldsymbol{\Phi}$) with the $L$ past values of the region's latent signal ($\mathbf{x}_m(t)$) and is represented mathematically by the vector inner product $\mathbf{b}_m\boldsymbol{\Phi}\mathbf{x}_m(t)$. Uncorrelated observation noise $e_m(t)$ with zero mean and variance $\sigma_m^2$ is then added to generate the observed signal $y_m(t)$. $e_m(t)$ is also assumed to be uncorrelated with $\mathbf{w}(t)$, at all $t$.

Equations (1–3) together represent a state-space model for estimating the causal interactions in latent signals based on observed multivariate fMRI time series. This model can be seen as an extension of GCA wherein VAR model for latent, rather than BOLD-fMRI, signals are used to model the causal interactions among brain regions. Furthermore, the MDSI model also takes into account variations in HRF while estimating causal interactions between the brain regions.

Estimating causal interactions between $M$ regions specified in the model is equivalent to estimating the parameter $\mathbf{C}_j$. In order to estimate $\mathbf{C}_j$ the other unknown parameters, $Q$, $\{\mathbf{b}_m\}_{m=1}^M$ and $\{\sigma_m^2\}_{m=1}^M$ and the latent signal $\{s(t)\}_{t=1}^T$ based on the observations $\{y_m^s(t)\}_{m=1,s=1}^{M,S}$, $t = 1, 2..T$, where $T$ is the total number of time samples and $S$ is number of subjects, needs to be estimated.

We modeled 2-back (high load) and 0-back (low-load) trials to estimate load-dependent causal interaction between brain networks. MDSI estimated strength of dynamic causal interaction per connection per task. A paired $t$-test was used to examine whether the strength of dynamic causal interaction between task conditions is different and multiple comparison correction was implemented using false discovery rate (FDR) correction ($p < 0.01$, Fig. 1).

**Identification of causal hubs.** We used the causal directed connectivity matrix estimated by MDSI to determine outflow and inflow hubs in each participant[13]. We computed the weighted node degree of each node by averaging outflow weights (all the input connections from a node to all other nodes) and then subtracting the average inflow weights (all the input connections to a node from all other nodes). Causal outflow and inflow hubs were identified as the nodes with the highest net positive, or net negative, node degree, respectively. This analysis was first conducted using MDSI-estimated connectivity matrices for the 2-back and 0-back conditions and then contrasting the 2- and 0-back conditions. We then determined whether outflow-inflow weights of each node in each condition were significantly different from zero and whether outflow-inflow weights of each node were significantly different between 2-back and 0-back conditions. Multiple comparisons were corrected using FDR ($p < 0.05$).

Supplementary Fig. 17 illustrates ROIs with their sizes proportionally scaled to their node degrees in a 3D brain and Supplementary Fig. 18 illustrates dynamic causal interactions between ROIs in a 3D brain.

**Modular structure of causal interactions.** We determined the community structure of dynamic causal interactions in the 2-back and 0-back working-memory task conditions using the Louvain algorithm[106], as implemented in the Brain Connectivity Toolbox[107]. The group-averaged context-dependent signed directed weighted connectivity matrices, estimated by MDSI, were entered into the analysis, with the resolution parameter gamma set to 1[108,109]. We used a consensus-based approach with 1000 iterations to handle potential degeneracy of community assignments[110]. In each iteration, a co-classification matrix in which each element contained 1 if two nodes were part of the same module and 0 otherwise was generated. We then averaged the resulting 1000 co-classification matrices to generate a co-occurrence matrix indicating the probability of two nodes being in the same module across 1000 iterations, which was then used to determine a consensus partition.

**Network controllability.** Dynamic control processes related to functional brain network organization were modeled as a linear, discrete, time-invariant systems of the form

$$\mathbf{x}(k+1) = \mathbf{A}\mathbf{x}(k) + \mathbf{B}\mathbf{u}(k) \qquad (4)$$

where $\mathbf{x}$ denotes the state vector, $\mathbf{A}$ is the weighted task condition-specific connectivity matrix ($\mathbf{C}_j$) estimated by MDSI, with elements of the matrix $\mathbf{A}$ describing causal directed connectivity between SN, FPN, and DMN nodes. The input matrix $\mathbf{B}$ identifies nodes controlled with input $\mathbf{u}$.

Classical results from control theory suggest that a system is controllable if and only if there exists a unique positive definite solution $\mathbf{W}$ to the "Lyapunov equation"

$$\mathbf{AW} + \mathbf{WA}' + \mathbf{BB}' = 0 \qquad (5)$$

The trace of controllability Gramian $\mathbf{W}_t$ was used as our quantitative metric of controllability[57] and the influence of each node on network dynamics. Details of our mathematical formulation are in the Supplementary Methods.

**Classification analysis.** To determine whether patterns of dynamic causal interactions can differentiate between 0-back and 2-back conditions, we conducted classification analysis using the linear support vector machine (SVM) algorithm from an open-source library, LIBSVM (for Library for Support Vector Machines; http://www.csie.ntu.edu.tw/cjlin/libsvm/). The MDSI weights used as a feature to predict 0-back or 2-back task conditions of each subject. The model was evaluated using the 10-fold cross-validation. Each time, one data fold was selected as a test set and the rest of the data were used as a training set. The training set was then used to train a SVM model, which was then applied to the test set for classification. This procedure was repeated 10 times with each data fold used exactly once as a test set. The significance of classification accuracy was evaluated using permutations (100 times).

**Brain-behavior analysis.** We applied canonical correlation analysis (CCA)[111] to explore the relation between dynamic causal interactions and working-memory performance. CCA is a statistical method for examining the relationships between two multivariate sets of variables, and has been shown to be a powerful tool for investigating brain-behavior relationships[112]. CCA finds the optimal linear combination of subjects' multivariate behavioral measures that maximize the relation between behavioral and brain measures. Specifically, brain features included dynamic causal interaction weights of connections in 0-back or 2-back conditions (110 connections in each task condition), and behavioral features included accuracy and RT in 0-back or 2-back conditions (1 score per condition). The significance of the canonical relationship was tested using Pillai's trace test with 5000 permutations.

**Stability analysis.** To evaluate robustness of MDSI findings, we conducted stability analysis using a bootstrapping procedure[46].

Stability analysis for multivariate dynamic causal interaction patterns was performed using the following steps:

(1) Randomly select a subset of samples from 737 participants without replacement.
(2) Apply $t$-test on each connection per condition or paired $t$-test on each connection between task conditions.
(3) Threshold dynamic causal interaction matrix at $p = 0.01$ (FDR correction).
(4) Repeat steps 1–4 500 times.
(5) Compute averaged thresholded dynamic causal interaction matrices.
(6) Compute the correlation between the averaged thresholded dynamic causal interaction matrix and that the matrix from the original full sample.
(7) Change subsample size from 20 to 600 and repeat steps (1)–(6). The subsample sizes include 20, 30, 40, 50, 60, 70, 80, 90, 100, 150, 200, 250, 300, 400, and 600.
(8) Generate stability graph by plotting correlation coefficients from step (6) over subsample size.

Stability analysis for the outflow hub of the rAI, the inflow hub of the rMFG, the load effect of network controllability, and high controllability in SN used the following steps:

(1) Randomly select a subset of samples from 737 participants without replacement.
(2) Apply paired $t$-test to determine whether the rAI has significantly higher outflow degree than other ROIs, whether the rMFG has significantly higher inflow degree than other ROIs, whether the controllability is significantly different between 2-back and 0-back across all the ROIs, and whether the controllability of the SN is significantly greater in 2-back than 0-back conditions ($p < 0.05$).
(3) Repeat steps (1) and (2) 500 times.
(4) Compute the probabilities that the rAI has significantly higher outflow degree than other ROIs, the rMFG has significantly higher inflow degree than other ROIs, the controllability is significantly different between 2-back and 0-back across all the ROIs, and the controllability of the SN is significantly greater in 2-back than 0-back conditions.
(5) Change subsample size from 20 to 600 and repeat steps (1)–(4). The subsample sizes include 20, 30, 40, 50, 60, 70, 80, 90, 100, 150, 200, 250, 300, 400, and 600.

(6) Generate stability graph by plotting the probabilities from step (5) over subsample size.

**Reporting summary**. Further information on research design is available in the Nature Research Reporting Summary linked to this article.

## Data availability

Source data are provided with this paper. The n-back working-memory task fMRI data is accessible from the HCP database (https://db.humanconnectome.org/). Simulation data are available in the Source data file. Source data are provided with this paper.

## Code availability

Functional MRI data preprocessing and statistical analyses were performed on the SPM12, FSL 6 (https://fsl.fmrib.ox.ac.uk/fsl/), Brain Connectivity Toolbox (https://sites.google.com/site/bctnet/), and Matlab 2020 (https://www.mathworks.com/products/matlab.html). MDSI, Network degrees and functional controllability scripts used in the study can be accessed at Github (https://github.com/scsnl/Cai_HCP_WM_MDSI_Controllability_2021) and Zenodo (https://doi.org/10.5281/zenodo.4706053)[113].

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

## Acknowledgements

We thank the Human Connectome Project (http://www.humanconnectome.org/) for making the data publicly available. The work is supported by grants from the National Institutes of Health MH105625 (W.C.), HD074652 (S.R.), EB022907 (V.M.), NS086085 (V.M.), and MH121069 (V.M. and W.C.), and Indo-US Science and Technology Forum (IUSSTF) IUSSTF/JC-110/2019 (R.P. and V.M.). We thank Drs. Yuan Zhang, Tianwen Chen, and Carlo de los Angeles for assistance with data analysis.

## Author contributions

Conceptualization: W.C. and V.M.; methodology: W.C., S.R., R.P., V.T., and V.M.; investigation: W.C., S.R., R.P., V.T., and V.M.; writing: W.C. and V.M.; review and editing: W.C., S.R., R.P., V.T., and V.M.; funding acquisition: W.C., S.R., and V.M.

## Competing interests

The authors declare no competing interests.
