## [Peer Review File · Nature Communications]

When text is deleted in rebuttals and referee reports, add “[redacted]” in that location.

Reviewer #1 (Remarks to the Author):

Summary of findings and general assessment:

The manuscript by Cai and co-authors presents an analysis of working memory-related dynamics among distributed brain areas and the modulation of dynamics by working memory load. The analyses were based on fMRI data of human subjects conducting an n-back memory task. ROIs were selected by either memory-load dependent activation (SN and FPN) or deactivation (DMN: PCC and VMPFC). Causal interaction was inferred from the ROI time series using Multivariate dynamical systems identification (MDSI), which models the hidden states that give rise to BOLD signals as a linear dynamical system. The transition matrix of hidden states is then inferred and interpreted as causal interaction.

With this causal interaction map, the authors then measured several network properties (community structure, net in/outflow, average controllability) and their load dependency at each node. The main findings were:

1. The community structure of the nodes is consistent with the prior division of ROIs (SN, FPN and DMN) and does not change with memory load.
2. Net outflow was the strongest in rAI (SN) and net inflow was the strongest in rMFG (FPN), regardless of memory load. Memory load dependency of net flow was highest in DMPFC (SN) among outflow nodes and in PCC (DMN) among inflow nodes.
3. Average controllability, defined as negatively related to the energy required to reach random states, was the highest in SN nodes and decreased with working memory load in general.
4. Canonical correlation analysis (CCA) between causal interaction and behavioral performance was significant in the 2-back condition but not in the 0-back condition. Causal interaction between SN and FPN positively correlated with performance, while causal interaction between DMN nodes negatively correlated with performance.

The authors conclude that the network properties such as causal (in/out flow) hubs and controllability change with working memory load and performance, revealing that SN and FPN demonstrate dissociable roles in cognitive system control.

This manuscript deals with the inter-areal dynamics that underly working memory and brings forward the concept of control theory as an analytical tool. In my opinion the topic is timely and of potential interest to many sub-fields within systems and cognitive neuroscience. However, I have several points that still need to be addressed.

Major points:

- 1) The validity of measuring average controllability using the connectivity matrix inferred from functional MRI still needs to be established. Controllability bears an operational and practical meaning, i.e. the possibility and feasibility (energy consumption) to reach certain states by e.g. injecting current into certain nodes. However, the connectivity inferred from task-fMRI data is conceptually a statistical dependency rather than a circuit diagram to be operated on (i.e. functional connectivity vs effective connectivity, Friston 2011 Brain Connectivity). Thus, there are at least two potential sources of error in this study:
 - a. Brain areas outside the ROIs and stimulus factors that are not accounted for in the state space model could be common causes of the reported functional connectivity (statistical dependency), making the analysis show “causal interaction” where in fact physiological connectivity does not exist. Brain areas outside the ROIs, although not the focus of the analyses, should be included in the state space model to ensure the accuracy of the connectivity among ROIs.
 - b. This is a bigger concern when comparing the controllability between different task conditions. In different task conditions, different nodes/neurons corresponding to that task condition are fixed at certain states. Depending on which nodes are fixed, it could either rightfully remove some false connectivity illustrated in a., if it is the common cause of the two nodes of the false connectivity, which can be generalized to any d-separation/conditional independence given these task conditions nodes; or it could falsely establish connectivity between the input nodes of the fixed task condition nodes (e.g. if a neuron is fixed at certain firing rate, its excitatory and inhibitory input will need to be balanced, showing correlation, and with a time delay, it will be detected as causality; collider pattern in a Bayesnet). Therefore, the difference of the inferred connectivity between 0-back and 2-back conditions is more likely to be functional/statistical connectivity rather

than effective/physiological connectivity.

If the authors assume it is functional connectivity that the controllability was measured on, this analysis - although found to be reliable with growing sample size and having the same mathematical formulation - would not mean the same as the controllability in an operational sense that relates to the ability to reach certain states (Gu et al., 2015 Nat Commun), because statistical connectivity doesn't ensure that a control input is passed on to a downstream node. To sum, the authors need to establish the validity of their analysis and elaborate on its interpretation. The assumptions and possible influences of the biases need to be clearly stated. I am stressing this because - to my knowledge - the application of controllability to a functional brain network is unprecedented.

2) Related to my previous point: a more elaborate and maybe graphical explanation of the meaning of controllability would help understanding. The current form of presentation is rather abstract. These statistics would mean barely more than just numbers to most audiences without looking into a series of other papers. The same applies to the concept of net flow, albeit to a lesser degree.

3) The ROIs are selected based on their modulation by memory load, which is also the main focus of the subsequent analyses. This raises the following questions: a) How much of the effects in the following analyses can already be explained by the activation level? b) Without cross-validation, how much of the effects are due to statistical double dipping? This is particularly relevant when reporting the performance of the machine learning algorithms. c) Are the results robust with respect to ROI selection? It would be helpful to do parallel analyses on the time series extracted using ICA or other un-supervised methods as in Taghia et al., 2018 Nat Commun.

4) The features used to predict memory load are very likely to be correlated (e.g. degree is the sum of connectivity, average controllability is highly correlated with unweighted degree, $r=0.91$). Would it be more meaningful and consistent to put features (e.g. degree, weights, controllability and raw activation) together in SVM and GLMnet and report their coefficients? This would be more representative of the contribution to predicting memory load and could reveal the non-redundant contribution. Comparison of nested models or step-wise entering of the features would also be good for e.g. showing the independent contribution of degree and controllability with respect to connectivity weights and raw activation.

Minor points:

5) The article takes a data-driven approach. To ensure an unbiased perspective of the data, the measures that are similar to average controllability and net flow also need to be reported.

a. The authors made claims about controllability but only average controllability was reported.

Average controllability is not representative of other forms of controllability such as modal controllability and boundary controllability (Gu et al., 2015 Nat Commun). Why is average controllability more relevant for working memory load? It would be important to compare these measures to gain a deeper understanding of which exact aspect changed with memory load.

b. Besides net flow, would other hub measures also show memory load modulation? What about the degree calculated by the undirected weights, or inflow and outflow separately instead of subtracting one from the other? It would also be informative to compare these measures in order to show for example in what exact ways SN and FPN are similar or different.

6) In the original paper reporting the MDSI method (Ryali et al., 2011 Neuroimage), the connectivity matrix is separated into intrinsic and modulatory components, and there is also a term that signifies external input. Why is the model in this paper simplified? Specifically, why are the two connectivity matrices in different memory loads not treated as modulatory connectivities of the same state space model?

7) In lines 144 and 145 (page 6), the authors argue that applying control theory on DTI data is ill-suited for assessing cognitive-context dependent effects and referred to reference [18], but reference [18] in fact discusses the limitation of certain techniques inferring connectivity from fMRI data, which lends no support to the authors' claim.

8) In line 148 (page 6) "to identify, for the first time, nodes that need the lowest energy to perturb the SN-FPN-DMN cognitive control system...", "for the first time" is overstated. See e.g. Cui et al., 2020 Elife.

9) The functional meaning of memory load is not really discussed in this article but rather treated as an abstract parameter. With more items in working memory, does it make the network more rigid, i.e. the states are more fixed, leading to a weaker coupling of the downstream areas and

thus reducing controllability? How are load-dependent statistics related to the number of items in working memory?

Conclusion:

In summary, this article deals with an important topic and touches a methodological frontier of applying control theory to task-fMRI inferred connectivity. This unprecedented analysis, however, requires more scrutiny and solidity. I would consider supporting publication if the authors are able to improve the manuscript in this regard in a revised version.

Reviewer #2 (Remarks to the Author):

Summary:

In this work, the authors applied causal interaction and network control theory to investigate the brain imaging during working memory tasks in the HCP dataset. They characterized the causal outflow and inflow hubs and examined the dynamic circuit mechanism with control theory. They found the dissociable roles of the SN and FPN in systems control and the asymmetry of control circuits operation during working memory. Overall this is an interesting work with novel perspective of investigating working memory circuits with control methods. The reasoning procedure can potentially be extended to understand other executive functions as well. But it still has many issues to address. I list my comments below.

Motivation & Introduction:

1. The current work wants to incorporate three concepts: causal inference, big data reproducibility, and brain control theory. However, from my understanding, these three parts are not coherently integrated. The authors need to better organize the material. My suggestion is that they first apply the causal inference to identify the hubs of information flow and areas encoding the individual difference; next, they utilize the network control theory to provide a mechanical explanation why the identified nodes play their roles in the circuits from the control perspective rather than simply report their differences in controllability measurement; finally, they could validate their findings based on the replicability.

2. I believe that the major contribution of the current work is the identification of the directed neural circuits underlining the working memory. So it seems to be inappropriate to firstly talk about the big-data approach in the second paragraph (line 74-80). In addition, they should introduce and discuss the limitation of the undirected network approach here since the working memory task has been investigated for a long time with traditional network approaches.

Results & Discussion:

1. To prove that the identified hubs are special for working memory tasks, they should at least include the results for resting state as well for exclusive purpose.

2. Need an explanation for why DMPFC and the PCC, rather than the AI and MFC are load-dependent.

3. Why do they group the nodes into systems to compare the controllability? Is there any relative difference across systems to differentiate with loads? For example, they could normalize the As for 0-back and 2-back scans to eliminate the effect of mean connection strength and prove that the findings were caused by network reconfiguration rather than increased/decreased connection caused by the variation of load.

4. Why do they use CCA to identify the dynamic causal interaction patterns? If the response is only 1-dimension or consists of simply accuracy and speed, I think linear regression with/without kernel was enough? (line 330-341).

5. They claimed that "We suggest that higher causal inflow into the MFG may facilitate maintenance and manipulation of the contents of working memory." If this is true, there should be observable differences by comparing loads both with and without normalization of connection matrices.

6. The explanation in line 464-466 for PCC's role in working memory is interesting. It would be better if this could be supported by related results in the context or additional statistical association or causal inferences.

7. What additional insight can we obtain when applying predictive models rather than comparing connection-wisely with FDR-FWER correction?

8. For the discussion in line 498-500, it would be better if they add additional explanation what types of reconfiguration cause the decrease of controllability for high load memory tasks.
9. For the discussion in 505-507, I am confused by the word "adequate" here. Since the effective control nodes are privileged to be hubs, is there any reason why SN stands out than FPN and DMN based on their nodal roles? Also, this should be supported by group comparison as well.
10. Figure 1A would be better if the node size was tuned proportionally to the nodal degree.
11. It would be great to have Figure 2B visualized on the brain as well.

Methods:

1. While the researchers are free to choose their ROIs accordingly, it would be good to use the full-brain model when they talk about the dynamics since the missing regions would also participate the signal progression unless they provide enough evidence that the regions not in the ROIs did not involve in the related information progression.
2. For the classification analysis (line 737-745), I don't think it is necessary to include the GLMNET results here. If so, they may want to compare the weight in the two models as well. In addition, a better setting from my understanding is to have 1/3 of data as test data and perform all experiments on 2/3 of the data. Then they could examine their results and models on the never-seen test data.
3. I am not sure whether CCA here is necessary. (line 754-764).
4. For the stability analysis for multivariate dynamic causal interaction patterns (line 770-784), I feel like in step(6) should be modified by reporting the distribution of correlation between each thresholded dynamic causal interaction matrix and the matrix from the left part rather than the pair of average matrix with original full data as the two parts were overlapped thus not fully proper for replicability test.
5. For the stability analysis for the outflow hubs of the rAI, the inflow hub of the rMFG, the load effect of network controllability and high controllability in SN (line 785-800), it seemed to be auxiliary with the statistical test values. Also, I would prefer to divide the dataset into three parts or train vs. test as above to prove the replicability.

Reviewer #3 (Remarks to the Author):

The present study reports network correlates of the visual n-back task in the Human Connectome Project (HCP) dataset, and their relationship to performance. The authors calculated effective connectivity among nodes in the salience (SN), frontoparietal (FPN), and default-mode (DMN) networks. The authors focused on two main metrics: asymmetry of directed influences and network controllability. The authors found a tendency for SN nodes to project outward with the anterior insula (AI) doing so in a static fashion, and the dorsomedial prefrontal cortex (DMPFC) doing so in a dynamic fashion, increasing its outward influence when cognitive demands increased. The FPN showed a more varied picture with what appears to be more static and dynamic outflow in the left hemisphere, and more static and dynamic inflow in the right hemisphere (*interpretation is my own here*). The various directed interactions were found to be related to performance in 2-back, but not 0-back. The authors also observed that the SN showed higher levels of network controllability, which diminished under high demands. Finally, the authors performed several robustness checks indicating sample sizes needed for stable estimates.

Overall, this is a beautiful set of analyses with some intriguing results. I appreciated not only the importance of what the authors are examining, but also the rigor with which they examined it. However, I was ultimately left wondering "what does it all mean?" As evidence, I typically finish my summary paragraphs (above) with a conclusion, but found myself unable to do so here. Clearly, there are some robust properties of the brain that have been estimated, and they are linked to cognitive behavior. But the manuscript seems to suffer from a lack of a strong theoretical framework. This makes it difficult to tie the results together, and determine how future work should build on these results. Thus, the lack of theory will limit the impact of this work. Therefore, I would suggest that the authors spend some time thinking through the theoretical implications of these data, sandwiching good theory around the beautiful results.

Here are some concrete examples of the theoretical shortcomings. In the introduction, the authors state:

“We hypothesized that causal interactions involving the SN, FPN, and DMN are dynamically modulated by working memory load and that directed network interaction patterns would differ between the high and low-load working memory conditions.”

I am not sure that anyone would hypothesize differently. Does anyone think that the brain is functionally the same during 0-back and 2-back? If some method that measures function found the brain to be the same across the two conditions, wouldn't one conclude that the measured is flawed? So, this does not setup for a very informative conclusion. But on the other end:

“We hypothesized that the AI node of the SN would emerge as a causal outflow hub during both the high and low working memory load conditions with significant inflow into FPN and DMN nodes.”

As far as I could tell, there was nothing in the reviewed literature that would lead to such a specific hypothesis. The reviewed literature focused on the network level, rather than the nodal level. This makes the reader wonder why the AI in particular? Why not the DMPFC or other parts of the SN? Why would this be a static pattern? Was the possibility that the AI could be an inflow hub not considered (it, after all, has a lot of afferent connections)? Hence, either there is more specific literature to be tied together or this is a rather peculiar case of Hypothesizing After the Results are Known (HARKing).

Then there is this:

“We further predicted that....the AI node of the SN would emerge as the node with high controllability.”

In this case, not only is this another oddly specific claim about one particular node, it also contradicts the study by Gu, Bassett, and colleagues (2015) that (to my knowledge) introduced controllability metrics to the neuroscience literature, which reported unremarkable controllability of the FPN and CO networks. There are notable differences among the studies (e.g. estimating controllability on fMRI vs DTI), but in the least, one would expect that this prior work would factor into the authors' hypotheses.

Collectively, I would encourage the authors to put as much effort into theory as they do with computation. If they do, I think there is strong potential for impact here. At the present, the manuscript reads as more of an archival report that would be suited for a more specialized outlet (e.g. NeuroImage). I personally think that would be a missed opportunity. But to seize the opportunity, the authors will need to be very clear regarding the mechanistic insight and theoretical advance that these data make.

Towards improving the theoretical framework, here are some specific thoughts that occurred to me while reading the manuscript. I want to make clear that it is not my intention that the authors should bend the manuscript to the way I see things – I believe it is inappropriate for reviewers to act as such. Nevertheless, I figured my specific thoughts might offer some food for thought that could facilitate a clearer theoretical framework. The authors should feel free to pick and choose which, if any, of the following they wish to incorporate:

- 1) N-back requires working memory, but studying n-back does not mean one is studying working memory. N-back is a complex task that requires many processes and a strong argument can be made that it is a cognitive control task more so than a working memory task. Consider that most contemporary working memory tasks find a means to isolate processes involved in maintenance/representation, while the focus on n-back is much more heavy on the control processes that act to update and manipulate representations. Hence, I think the authors start off on the wrong foot by describing working memory. Most working memory experts that I have had discussions with agree that there is very little that the HCP n-back data can do to advance our knowledge of working memory.

- 2) What these data and analyses are well-positioned to do is tell us about control. All of the

analyses that the authors utilize are about control. Asymmetries of directed influence are the essence of how one brain region controls another. A critical failing of much work on control is that it fails to measure such directed influences. Most work uses functional connectivity, which is non-directional, or structural connectivity, which not only fails to have direction, but also fails to have any dynamics (works by Cole, Bassett, Bertolero, Gratton, and others come to mind). Therefore, the authors have measures that can offer insights into control that other methods cannot. In my opinion, this is what sets this work apart and I believe it fits more naturally in a control setting rather than a working memory setting.

3) From a control perspective, the SN and FPN are obviously important networks. However, their relative order in a hierarchical sense is unknown. As far back as classic work by Botvinick and colleagues, the SN (DMPFC in particular), has been thought to drive the FPN (DLPFC in particular). To my knowledge, such directed relationships have never been convincingly established. On the other hand, the FPN is literally referred to as the "control network". Interestingly, work on hierarchical control has often ignored the SN in favor of the FPN in isolation (e.g. Koechlin, Badre, D'Esposito, Nee) despite the theoretical importance of the SN. So, having two different methods (asymmetries of directed interactions, and network controllability) that can examine the relative control properties of the SN and FPN can potentially bring to light the importance of the SN in driving control.

4) Most work on control either treats both hemispheres of the FPN as homogenous or ignores a hemisphere (often ignoring the right...c.f. works by Badre, Nee, D'Esposito, others). However, the frontal and parietal lobes are not symmetrical, in part due to the language network's presence in the left. A striking aspect of this work is that the right and left FPN clustered into distinct communities with very different inflow/outflow properties. The authors don't seem to key in on this point, but it has important implications for future work. As the authors are aware, right-sided frontal parietal areas are most often implicated in control over responses (e.g. stop signal task). In a hierarchical control sense, responses are low in the hierarchy. Then, I wonder whether these data suggest a control hierarchy that goes SN -> left FPN -> right FPN from motivation (SN) to abstract control (left FPN) to concrete control (right FPN). A speculative possibility that likely requires some further thought and scholarship. But it is this sort of advancing of theory that I think would really push the manuscript to the level of making a substantial impact.

Again, I stress that these are my thoughts colored by my own work, but it's not my manuscript and I certainly would not hold the authors to my perspective. Any way that the authors choose, it just needs to be made clear how theory can advance in light of these data.

One final thought: I greatly appreciate the analyses the authors performed demonstrating the robustness of their metrics. However, it may be misleading to suggest that sample size is the primary driver of robustness of network level metrics. The Midnight Scan Club data make it clear that one does not need a large sample sizes to get robust measures. Nee, 2019, *Communications Biology* also makes the point that perhaps the HCP is not representative given the unusually short task scans. As Nee was cited as among the studies using small samples with "inconsistent and contradictory findings", the authors might be interested in a recent preprint on bioRxiv on that same "small sample" dataset showing excellent consistency of DCM parameters indicating that sample size is not necessary for such robustness (<https://www.biorxiv.org/content/10.1101/2020.03.30.016394v1>). Big data has its own set of flaws, and I would argue that much of, if not most progress in neuroscience has come from carefully collected small samples (consider e.g. animal neurophys, brain damaged patients). Hence, while I absolutely agree that robustness is vital, I believe it is misleading to suggest that large samples are the best way to get it.

We thank the reviewers for their valuable feedback and constructive suggestions: “this is a beautiful set of analyses with some intriguing results. I appreciated not only the importance of what the authors are examining, but also the rigor with which they examined it.” We have clarified all issues raised by the reviewers and have provided detailed response to each of their comments. We have also conducted extensive additional analyses and updated our manuscript with detailed results from these analyses.

Reviewer #1 (Remarks to the Author):

Summary of findings and general assessment:

The manuscript by Cai and co-authors presents an analysis of working memory-related dynamics among distributed brain areas and the modulation of dynamics by working memory load. The analyses were based on fMRI data of human subjects conducting an n-back memory task. ROIs were selected by either memory-load dependent activation (SN and FPN) or deactivation (DMN: PCC and VMPFC). Causal interaction was inferred from the ROI time series using Multivariate dynamical systems identification (MDSI), which models the hidden states that give rise to BOLD signals as a linear dynamical system. The transition matrix of hidden states is then inferred and interpreted as causal interaction.

With this causal interaction map, the authors then measured several network properties (community structure, net in/outflow, average controllability) and their load dependency at each node. The main findings were:

- 1. The community structure of the nodes is consistent with the prior division of ROIs (SN, FPN and DMN) and does not change with memory load.*
- 2. Net outflow was the strongest in rAI (SN) and net inflow was the strongest in rMFG (FPN), regardless of memory load. Memory load dependency of net flow was highest in DMPFC (SN) among outflow nodes and in PCC (DMN) among inflow nodes.*
- 3. Average controllability, defined as negatively related to the energy required to reach random states, was the highest in SN nodes and decreased with working memory load in general.*
- 4. Canonical correlation analysis (CCA) between causal interaction and behavioral performance was significant in the 2-back condition but not in the 0-back condition. Causal interaction between SN and FPN positively correlated with performance, while causal interaction between DMN nodes negatively correlated with performance.*

The authors conclude that the network properties such as causal (in/out flow) hubs and controllability change with working memory load and performance, revealing that SN and FPN demonstrate dissociable roles in cognitive system control.

This manuscript deals with the inter-areal dynamics that underly working memory and brings forward the concept of control theory as an analytical tool. In my opinion the topic is timely and of potential interest to many sub-fields within systems and cognitive neuroscience. However, I have several points that still need to be addressed.

Major points:

1.1 The validity of measuring average controllability using the connectivity matrix inferred from functional MRI still needs to be established. Controllability bears an operational and practical meaning, i.e. the possibility and feasibility (energy consumption) to reach certain states by e.g. injecting current into certain nodes. However, the connectivity inferred from task-fMRI data is conceptually a statistical dependency rather than a circuit diagram to be operated on (i.e. functional connectivity vs effective connectivity, Friston 2011 Brain Connectivity). Thus, there are at least two potential sources of error in this study:

- a. Brain areas outside the ROIs and stimulus factors that are not accounted for in the state space model*

could be common causes of the reported functional connectivity (statistical dependency), making the analysis show “causal interaction” where in fact physiological connectivity does not exist. Brain areas outside the ROIs, although not the focus of the analyses, should be included in the state space model to ensure the accuracy of the connectivity among ROIs.

b. This is a bigger concern when comparing the controllability between different task conditions. In different task conditions, different nodes/neurons corresponding to that task condition are fixed at certain states. Depending on which nodes are fixed, it could either rightfully remove some false connectivity illustrated in a., if it is the common cause of the two nodes of the false connectivity, which can be generalized to any d -separation/conditional independence given these task conditions nodes; or it could falsely establish connectivity between the input nodes of the fixed task condition nodes (e.g. if a neuron is fixed at certain firing rate, its excitatory and inhibitory input will need to be balanced, showing correlation, and with a time delay, it will be detected as causality; collider pattern in a Bayesnet).

Therefore, the difference of the inferred connectivity between 0-back and 2-back conditions is more likely to be functional/statistical connectivity rather than effective/physiological connectivity.

If the authors assume it is functional connectivity that the controllability was measured on, this analysis - although found to be reliable with growing sample size and having the same mathematical formulation - would not mean the same as the controllability in an operational sense that relates to the ability to reach certain states (Gu et al., 2015 Nat Commun), because statistical connectivity doesn't ensure that a control input is passed on to a downstream node. To sum, the authors need to establish the validity of their analysis and elaborate on its interpretation. The assumptions and possible influences of the biases need to be clearly stated. I am stressing this because - to my knowledge - the application of controllability to a functional brain network is unprecedented.

Response: Unlike functional and effective connectivity, Bayesnet or GCA, MDSI (like DCM) estimates condition specific causal interactions in the latent neuronal space rather than the observed fMRI. This is done explicitly to minimize the influences of regional variation in HRF which can confound dynamic causal analysis of fMRI data. Currently, MDSI (and related techniques such as DCM) cannot be scaled up to include areas spanning the entire human brain. This tradeoff is necessary to estimate physiologically meaningful causal interactions. The particular ROIs chosen in this study are based on our systems-neuroscience based theoretical models involving key nodes of the SN, CEN and DMN. Moreover, MDSI estimates the full asymmetric connectivity matrix without the need for sparsification and testing exponentially large number of models. To address the concerns raised about ROI selection we have conducted several additional analyses using alternate approaches to extracting ROIs from SN, CEN and DMN (see Response 1.3). These analyses replicated our original findings.

Even outside the context of state-space models which can take regional variations in HRF into account, estimating functional connectivity with methods other than simply computing correlation coefficients is challenging. For example, while partial correlation matrix can reveal direct interactions between brain regions, applying at the whole brain level is hard due to the non-invertibility of the correlation matrix^{1,2}. Furthermore, the resulting connectivity graph structure is still symmetric. As noted by Reviewer 3, “Asymmetries of directed influence are the essence of how one brain region controls another”, and this is the crucial difference in the approach taken by our study. Here we investigate task-based asymmetric causal connectivity matrices estimated in the latent neuronal space using standard space-space models³, albeit tailored to handle the complexities of inter-regional variations in fMRI data.

The notion of “reaching specific states” based on structural brain measures is problematic as brain states are unknown and ill-specified, notwithstanding previous claims. In the case of task-fMRI based studies the link is a little more direct in that individual task conditions are associated with distinct brain and mental states. More generally, the question of control of states in complex system remains an open unresolved issue because only a fraction of the state variables are directly measurable even for much simpler systems⁴.

Furthermore, a crucial problem is that structural topology-based approaches are not good proxies for modeling controllability in dynamical systems ⁵. Using multiple examples, Leitold et al. ⁵ have drawn attention to the importance of dynamics between state variables for the study of controllability in complex systems. The resulting (functional) networks differ from structural topologies of the system and describe more accurately state space dynamics and controllability ⁶. We take a similar approach here and in a two-step process determine causal links associated with the underlying state space dynamics and then use it to determine controllability. Thus, our approach is similar to the theoretical approach advocated by Leitold et al. ⁵. Future work will benefit from integrating structural topology with functional measures, but this is beyond the scope of the present study given challenges with estimating DTI fiber tracks between the focal SN, CEN and DMN ROIs examined in the present study.

1.2 Related to my previous point: a more elaborate and maybe graphical explanation of the meaning of controllability would help understanding. The current form of presentation is rather abstract. These statistics would mean barely more than just numbers to most audiences without looking into a series of other papers. The same applies to the concept of net flow, albeit to a lesser degree.

Response: We thank the reviewer for this suggestion and have now addressed this issue at length in the revised manuscript. We have elaborated on the theoretical basis of our use of average controllability and its relation to network structure in asymmetric dynamical systems models. In the revised Supplementary Methods and Results, we have also now clarified our methods and approach with additional simulations, highlighting some new results in the process.

Average Controllability

Dynamic control processes are modeled as a linear, discrete, time invariant systems of the form

$$x(k + 1) = Ax(k) + Bu(k)$$

$x \in R^n$ denotes the state of the various brain regions over time. The matrix $A \in R^{n \times n}$ is the system matrix (which in our case is a weighted connectivity matrix) whose elements describe interactions between individual states. The input matrix B identifies the control input vector and is usually of the form $B = [e_1 \dots e_m]$, e_i denotes the n -dimensional i -th canonical vector. $u(k)$ is the control signal.

Controllability means the ability of the system to transfer from a given initial state to any target state, in finite time, by means of an external control input. In the context of brain networks it means the ability to drive the brain to specific states that facilitate adaptive behaviors and cognition.

Classical results from Control Theory suggest that a system is controllable if and only if there exists a unique positive definite solution W to the “Lyapunov equation” ⁷

$$AWA' - W + BB' = 0$$

In particular the unique positive solution takes the form

$$W = \sum_{\tau=0}^{\infty} A^{\tau} B B' (A')^{\tau}$$

This positive definite matrix W is referred to as the Controllability Gramian and the trace of the controllability Gramian can be used to compute a quantitative measure of the influence of each node over the entire network. This quantity is termed as the average controllability and is inversely related to the average control energy. Nodes or modules with high average controllability are the ones which are

expected to have a larger (control-) influence over the network. In terms of the energy a large W would mean a smaller amount of energy required to transfer the system to any point in the state space.

As a simple illustration consider a causal directed graph with 5 nodes as shown in **Figure 1**. For simplicity we assume that the input signal u is applied to node 1. Controllability of the network via this input would mean the ability of the node where input of applied (node 1 in this case) to influence (or control) the other nodes (2-5). As we show below, the energy needed to perturb a specific state is directly related to the weighted sum of the out degree. In general, if there is no directed path between node i and node j , then the node j cannot be controlled or influenced by a control input at node i and hence the network is said to be uncontrollable from node i . The network in this example can be controlled by input to any of the 5 nodes. For a complete graph theoretical explanation, we refer to ⁵.

Figure 1. 5 node network illustrating that input u to node 1 can control all other network nodes.

Average controllability is related to weighted out-degree in asymmetric networks

Current work on controllability in human brain networks is based on symmetric connections estimated using DTI. However, the connectivity matrix need not be symmetric even in the case of structural connectivity: it is well known that most anatomical connections in the brain are not symmetric ⁸, which poses other problems for the application of DTI-based measures in controllability analysis. Crucially, for our purposes here, we show below that under more general conditions in which the connectivity matrix is not symmetric, average controllability is related to the weighted out-degree but not the weighted in-degree or the weighted degree. To clarify this, we generated a random 50-node network and examined average controllability in relation to several node degree measures.

We evaluated the relation between average controllability and weighted out-degree. We evaluated the weighted out-degree as (a) the algebraic sum of the outgoing edge weights for each node: $d_i = \sum_{j=1}^n a_{ji}$. (b) sum of the absolute weight of the causal influences: $|d_i| = \sum_{j=1}^n |a_{ji}|$. We found that the average controllability was related to the absolute weighted out-degree but not the signed out-degree (**Figure 2**).

Next, we conducted similar analyses examining average controllability vs in-degree. We evaluated the weighted out-degree as (a) the algebraic sum of the incoming edge weights for each node: $d_i = \sum_{j=1}^n a_{ij}$. (b) sum of the absolute weighted incoming causal influences: $|d_i| = \sum_{j=1}^n |a_{ij}|$. We found that the average controllability was not related to either the absolute weighted in-degree or the signed in-degree (**Figure 3**).

We then evaluated the relation between average controllability and the difference in the absolute values of the weighted out vs. in degree: $|dI_i| = |\sum_{j=1}^n |a_{ji}| - \sum_{i=1}^n |a_{ij}| |$. We found that average controllability was not correlated with the net difference (**Figure 4**).

These analyses demonstrate that average controllability depends crucially on network asymmetries, and are consistent with the intuition that nodes with the highest outflow also have the highest ability to modulate ongoing neural activity in other brain regions.

Finally, we generalized these analyses using 100 random networks of 100 nodes. As in the example above, average controllability was only correlated with absolute sum of the weighted out-degree (**Figure 5**), again.

Figure 2. Average controllability is (left) not correlated with the weighted out-degree, but (right) is strongly correlated with the absolute weighted out-degree.

Figure 3. Average controllability is (left) not correlated with the weighted in-degree, nor (right) the absolute weighted in-degree.

Figure 4. Average controllability is not correlated with the difference in absolute values of the weighted out vs. in degree.

Figure 5. Averaged controllability vs. network degree across 100 random networks with 11 nodes. 1: weighted-out degree, 2: absolute weighted-out degree, 3, 4: same with in-degree, 5: difference between absolute weighted out – in degree.

1.3 The ROIs are selected based on their modulation by memory load, which is also the main focus of the subsequent analyses. This raises the following questions: a) How much of the effects in the following analyses can already be explained by the activation level? b) Without cross-validation, how much of the

effects are due to statistical double dipping? This is particularly relevant when reporting the performance of the machine learning algorithms. c) Are the results robust with respect to ROI selection? It would be helpful to do parallel analyses on the time series extracted using ICA or other un-supervised methods as in Taghia et al., 2018 Nat Commun.

Response: We have conducted additional analyses along the lines suggested by the reviewer. These analyses replicated our original results and demonstrate the robustness of our findings. These new results are described in page 18 and in detail in the revised Supplementary Materials. Several noteworthy points emerged from this analysis. First, although ROIs were determined based on memory load, load modulation of activation was not directly associated with load modulation of dynamic causal interaction. Indeed, only the lIPL showed a weak, but significant, correlation between load modulation of activation and net causal influence (outflow – inflow degree) ($r=0.11$, $p=0.003$). None of the other ROIs showed significant correlation between load modulation of regional activation and net causal influence (all $ps>0.05$).

Second, our classification model was evaluated using 10-fold cross validation. In this procedure, one data fold was selected as a test set and the rest of the data were used as a training set. The training set was then used to train an SVM model, which was then applied for classification in the test set. This procedure was repeated 10 times with each data fold used exactly once as a test set. The significance of classification accuracy was evaluated using permutations (100 times). Furthermore, in response to the reviewer #2's comment 2.18, we split the data into three subsets and replicated all the findings in each subset, which demonstrates the robustness of our findings (see Response 2.18 for details).

Third, to further demonstrate the reproducibility of our findings, we conducted additional analysis using another set of ROIs determined by meta-analysis of working memory studies. Specifically, we searched “working memory” term in NeuroSynth (<https://www.neurosynth.org/>), which identified 1,091 studies and 39,905 activations. We selected activation peaks in the lAI, rAI, DMPFC, lMFG, lFEF, lIPL, rMFG, rFEF and rIPL. Because the meta-analysis did not report deactivations, we kept the original PCC and VMPFC ROIs. Each ROI was created using 6-mm radius spheres whose centers are activation peaks. We repeated our analyses and reported main findings in the Supplementary material and summarized below:

MDSI identified several links that showed significant dynamic causal interactions in the 2-back and 0-back task conditions ($p < 0.01$, FDR-corrected) (**Figure 6A**). Next, leveraging the large sample size of the HCP dataset, we examined the stability of dynamic causal interaction patterns using bootstrapping with subsamples ranging from 20 to 600. We found that dynamic causal interaction patterns achieved a high level of stability ($r > 0.8$) with subsample sizes of $N=30$ or more (**Figure 6B**). These results demonstrate that MDSI reliably estimates dynamic causal interaction patterns associated with both the 2-back and 0-back conditions.

A. Directed causal influences

B. Stability of causal influences

Figure 6. Analysis using ROIs determined from NeuroSynth meta-analysis. **(A)** Significant directed causal influences between SN, FPN and DMN ROIs in the 2-back and 0-back working memory task conditions ($p < 0.01$, FDR corrected). Red cells indicate significant positive influences and blue indicates significant negative influences. **(B)** Stability analyses revealed highly stable multivariate patterns of causal influences among SN, FPN and DMN nodes in 2-back and 0-back task conditions ($r > 0.8$ for Sample size > 25). X-axis shows sample sizes ranging from 20 to 600. Y-axis shows stability, computed as the correlation of multivariate causal influence patterns between the original sample and random subsamples drawn from $N=737$ participants.

We then computed the outflow degree of each node in each task and participant. The outflow degree is the weighted node degree: averaged outflow weights (all the output connections from a node to all other nodes) *minus* averaged inflow weights (all the input connections to a node from all other nodes). The rAI and lAI showed significant positive outflow in both the 2-back and 0-back conditions, with the rAI showing the highest outflow degree ($p < 0.05$, FDR corrected, **Figure 7A**). Stability analysis revealed that this rAI finding was highly stable ($> 80\%$) with sample sizes of $N=200$ or more (**Figure 7B**). That is, the rAI showed the consistently highest outflow, across multiple random subsamples of the data.

Similar to our original analysis, the rMFG showed significant inflow ($p < 0.05$, FDR corrected, **Figure 7A**). Stability analysis also revealed that this rMFG finding was highly reliable ($> 80\%$) with subsample sizes of $N=200$ or more (**Figure 7B**). That is, the rMFG showed the consistently highest inflow, across multiple random subsamples of the data.

A. Directed causal outflow

B. Stability of causal outflow

Figure 7. Analysis using ROIs determined from NeuroSynth meta-analysis. **(A)** AI showed the highest directed causal outflow between SN, FPN and DMN nodes in both the 2-back and 0-back working memory task conditions. In contrast, the rMFG showed the highest directed causal inflow among all nodes in both task conditions. **(B)** Stability analyses revealed highly stable directed causal outflow from the rAI and directed causal inflow into the rMFG in both the 2-back and 0-back working memory task conditions. X-axis shows sample size, ranging from 20 to 600. Y-axis shows stability, computed as the probability that the rAI shows the highest positive directed causal outflow among SN, FPN and DMN nodes, and the probability that the rMFG shows the highest causal inflow in random subsamples drawn from $N=737$ participants.

We next examined whether multivariate patterns of dynamic causal interactions differed between the two task conditions. A support vector machine (SVM) algorithm with 10-fold cross-validation revealed a classification accuracy of 68% ($p < 0.01$, permutation test, **Figure 8**). To replicate these findings, we then used Lasso and Elastic-Net Regularized Generalized Linear Models (GLMNET) with a 10-fold cross validation. GLMNET analysis revealed a classification accuracy of 68% ($p < 0.01$, permutation test, **Figure 8**).

Figure 8. Analysis using ROIs determined from NeuroSynth meta-analysis. Both linear SVM and GLMNET analyses with 10-fold cross validation revealed that dynamic causal influences between SN, FPN and DMN nodes distinguished the 2-back and 0-back working memory task conditions ($ps < 0.01$).

Next, we sought to determine specific links which differ in the strength of dynamic causal interactions between the 2-back and 0-back conditions (all $ps < 0.01$, FDR corrected; **Figure 9A**). Increased dynamic causal interactions in the 2-back condition were observed primarily between SN and FPN nodes. In contrast, dynamic causal influences from the PCC node in the DMN on the SN and FPN decreased in the 2-back, compared to the 0-back, condition.

We further examined the stability of working-memory load dependent dynamic causal interaction patterns. We found that dynamic causal interaction patterns achieved a high level of stability ($r > 0.8$) with sample sizes of $N=200$ or more (**Figure 9C**). These results demonstrate that MDSI reliably estimates dynamic causal interaction patterns associated with working memory-load.

We then contrasted the net causal influences of each node between the high and low-load working memory conditions. The lIPL showed significantly greater outflow in the 2-back, compared to the 0-back, task condition ($p < 0.05$, FDR corrected; **Figure 9B**) and the DMPFC had marginally significantly greater outflow in the 2-back than 0-back condition ($p=0.08$), whereas the PCC showed significantly higher net inflow in the 2-back, compared to the 0-back, condition ($p < 0.05$, FDR corrected; **Figure 9B**).

A. Directed causal influences

B. Directed causal outflow

C. Stability of causal influences

Figure 9. Analysis using ROIs determined from NeuroSynth meta-analysis. **(A)** MDSI analysis revealed links with significantly greater directed causal influences between SN, FPN and DMN nodes in the 2-back, compared to the 0-back, working memory task condition ($p < 0.05$, FDR corrected). **(B)** The IAI, DMPFC and IPL showed significantly higher directed causal outflow in the 2-back, compared to the 0-back, task condition. In contrast, the PCC showed significantly higher directed causal inflow in the 2-back, compared to the 0-back, task condition ($p < 0.05$, FDR corrected). **(C)** Stability analyses revealed highly stable multivariate patterns of causal influences between SN, FPN and DMN nodes in 2-back versus 0-back ($r > 0.8$ with samples > 200). X-axis is the subsample sizes, ranging from 20 to 600. Y-axis is the stability measures, which is the correlation of multivariate causal interaction patterns between subsamples and original dataset.

Network controllability was evaluated for each node and task condition and entered into an ANOVA with factors working memory load and node. We found a significant main effect of node ($F_{10,736} = 45.16$, $p <$

2.0e-16) and load ($F_{1,736} = 24.49, p < 9.25e-07$) (**Figure 10A**). Network controllability was lower in the 2-back, compared to the 0-back condition, and this finding held for all nodes (all $ps < 0.001$).

To further evaluate the differential controllability of the three brain networks, we grouped ROIs' controllability scores by their networks and conducted an ANOVA with factors working memory load and network (SN, FPN and DMN). We found a significant main effect of network ($F_{2,736} = 213.7, p < 2.0e-16$), and load ($F_{1,736} = 24.27, p < 1.03e-6$), and a significant interaction between load and network ($F_{2,736} = 6.95, p < 0.001$). SN nodes (rAI, lAI, and DMPFC) had the highest level of controllability in both the 0-back and 2-back tasks. Network controllability was higher in the 0-back, compared to the 2-back condition, and this finding held for all three networks (all $ps < 0.001$, Bonferroni corrected) (**Figure 10B**). Further analysis revealed that the load x network interaction arose from load differences characterized by higher controllability of the SN compared to the FPN ($t = 1.89, p = 0.05$) and the DMN ($t = 3.23, p = 0.001$), and higher controllability of the FPN compared to DMN ($t = 2.17, p = 0.03$).

We then examined the stability of these findings, focusing first on working-memory load dependent differences in network controllability. Higher network controllability on the 0-back, compared to the 2-back, condition achieved a high level of stability (>80%), with sample sizes of N=200 or more (**Figure 10C**). Results showing network differences, with SN having the highest network controllability, also showed a high level of stability (>80%) with samples of N=100 or more for the 2-back task condition and N=150 or more for the 0-back condition.

A. Node-level controllability

B. Network-level controllability

C. Stability of controllability

Figure 10 Analysis using ROIs determined from NeuroSynth meta-analysis. **(A)** Functional network controllability, assessed in each brain node, was significantly lower in the 2-back, compared to the 0-back, working memory task condition ($p < 0.001$). AI and DMPFC nodes in the SN have significantly higher controllability than FPN and DMN nodes, except for IFEF and IMFG nodes of the FPN in the 2-back task condition and IFEF in the 0-back task condition ($p < 0.001$). **(B)** Functional network controllability, assessed across SN, FPN and DMN nodes, was significantly lower in the 2-back,

compared to the 0-back, working memory task condition ($p < 0.001$). The SN shows significantly higher controllability than the FPN and DMN ($p < 0.001$). (C) Stability analyses revealed stable load effect (0-back > 2-back) and network difference (SN > FPN and SN > DMN). X-axis shows sample size, ranging from 20 to 600. Y-axis shows stability, computed as the probability that the load effect of controllability is significantly different between 2-back and 0-back working memory conditions, and the probability that the SN shows greater controllability than the FPN and DMN in both 2-back and 0-back working memory conditions, in random subsamples drawn from $N=737$ participants.

Finally, we investigated whether dynamic causal interactions between the SN, FPN and DMN are related to working memory performance. CCA model fits were significant in the 2-back condition (*Pillai's trace* = 0.37, $p = 0.004$) but not in the 0-back condition (*Pillai's trace* = 0.28, $p = 0.7$). CCA identified a significant relation between dynamic causal weights and behavioral scores in the 2-back condition ($r = 0.47$, $p < 0.001$, **Figure 11A**). **Figure 11B** illustrates the canonical correlation coefficients and highlights positive influences between SN and FPN nodes and negative influences of SN and FPN nodes on PCC and VMPFC nodes of the DMN.

Figure 11. Analysis using ROIs determined from NeuroSynth meta-analysis. (a) Canonical correlation analysis revealed a significant relationship between directed SN, FPN and DMN causal influences and behavioral performance in the 2-back working memory task condition ($r = 0.47$, $p < 0.001$). (b) Correlation coefficients contributing to brain-behavior relations highlights positive influences between SN and FPN nodes and negative influences of SN and FPN nodes on PCC and VMPFC nodes of the DMN.

In summary, these results demonstrate that the entire set of originally reported findings were replicated with the new set of ROIs, the only exception was that between-condition differences in net causal influence from the revised DMPFC node was relatively weaker.

1.4 The features used to predict memory load are very likely to be correlated (e.g. degree is the sum of connectivity, average controllability is highly correlated with unweighted degree, $r=0.91$). Would it be more meaningful and consistent to put features (e.g. degree, weights, controllability and raw activation) together in SVM and GLMnet and report their coefficients? This would be more representative of the contribution to predicting memory load and could reveal the non-redundant contribution. Comparison of nested models or step-wise entering of the features would also be good for e.g. showing the independent contribution of degree and controllability with respect to connectivity weights and raw activation.

Response: Our goal was to test the hypothesis that dynamic causal interaction patterns differentiate 2- and 0-back working memory conditions. To address the reviewer's question, we conducted additional

analyses. We examined whether each type of node-wide brain metrics, e.g. raw activation, node degree and controllability, could distinguish task conditions. Classification analyses using Glmnet showed significant cross-validation accuracy based on raw activation (ACC=81%, $p<0.01$, permutation test), on node degree (ACC=56%, $p<0.01$, permutation test) and on controllability (ACC=56%, $p<0.01$, permutation test). Each Glmnet model's coefficients are summarized in **Table 1**. Finally, we examined classification performance using combined features from raw activation, node degree and controllability. Glmnet analysis showed significant cross-validation accuracy (ACC=82%, $p<0.01$, permutation test).

Table 1. Coefficients of each Glmnet model in differentiating 2-back and 0-back task conditions based on raw activations, node degree and controllability of ROIs.

ROIs	Activation	Node Degree	Controllability
lAI	0.000	0.000	0.000
rAI	0.000	-0.024	-0.231
DMPFC	0.015	0.020	0.000
lMFG	0.026	0.023	0.000
lFEF	0.066	0.011	0.000
lIPL	0.022	0.021	0.022
rMFG	0.029	-0.001	0.000
rFEF	0.099	-0.032	0.000
rIPL	0.087	-0.028	0.144
PCC	-0.083	-0.045	0.000
VMPFC	-0.080	-0.019	0.000

1.5 The article takes a data-driven approach. To ensure an unbiased perspective of the data, the measures that are similar to average controllability and net flow also need to be reported.

a. The authors made claims about controllability but only average controllability was reported. Average controllability is not representative of other forms of controllability such as modal controllability and boundary controllability (Gu et al., 2015 Nat Commun). Why is average controllability more relevant for working memory load? It would be important to compare these measures to gain a deeper understanding of which exact aspect changed with memory load.

[redacted]

[redacted]

[redacted]

[redacted]

[redacted]

[redacted]

[redacted]

1.6 In the original paper reporting the MDSI method (Ryali et al., 2011 Neuroimage), the connectivity matrix is separated into intrinsic and modulatory components, and there is also a term that signifies external input. Why is the model in this paper simplified? Specifically, why are the two connectivity matrices in different memory loads not treated as modulatory connectivities of the same state space model?

Response: The models we used in this manuscript and that proposed in Ryali et al.,¹⁵ are the same. The model used here is just a reparameterization of that in Ryali et al.,¹⁵ which we will show here. The model used earlier was

$$s(t) = As(t - 1) + \sum_{j=1}^J v_j(t)B_j s(t - 1) + w(t) \quad (1)$$

Where, A is the intrinsic and B_j 's are the intrinsic and modulatory connectivity matrices respectively. $v_j(t), j = 1$ to J are the experimental conditions. We know that

$$\sum_{j=1}^J v_j(t) = 1$$

Therefore, (1) can be written as:

$$\begin{aligned} s(t) &= \sum_{j=1}^J v_j(t) As(t - 1) + \sum_{j=1}^J v_j(t)B_j s(t - 1) + w(t) \\ s(t) &= \sum_{j=1}^J v_j(t) [A + B_j]s(t - 1) + w(t) \\ s(t) &= \sum_{j=1}^J v_j(t) C_j s(t - 1) + w(t) \quad (2) \end{aligned}$$

Where, $C_j = A + B_j, j = 1, 2, \dots, J$. We use the model (2) in this work which is just a reparameterization of the original model (1). We use the parameterization in (2) because the condition specific causal connectivity matrix is more reliable to estimate than estimating the modulatory component B_j which is equal to $C_j - A$, the difference between the condition specific and intrinsic connectivity matrices. Formulation in (2) is not a simplification of the original model (1). The condition specific causal connection matrices (C_j 's) are estimated within the same state space modeling framework. We did not explicitly model the influence of external stimulus because of identifiability issues, instead this is implicit in our modeling of working memory load-specific causal interactions.

1.7 In lines 144 and 145 (page 6), the authors argue that applying control theory on DTI data is ill-suited for assessing cognitive-context dependent effects and referred to reference [18], but reference [18] in fact discusses the limitation of certain techniques inferring connectivity from fMRI data, which lends no support to the authors' claim.

Response: We apologize for this error and have changed to the correct reference (Tu C et al. 2018) in the revised manuscript.

1.8 In line 148 (page 6) “to identify, for the first time, nodes that need the lowest energy to perturb the SN-FPN-DMN cognitive control system...”, “for the first time” is overstated. See e.g. Cui et al., 2020 Elife.

Response: As suggested by the reviewer, we have removed “for the first time” from the revised manuscript.

1.9 The functional meaning of memory load is not really discussed in this article but rather treated as an abstract parameter. With more items in working memory, does it make the network more rigid, i.e. the states are more fixed, leading to a weaker coupling of the downstream areas and thus reducing controllability? How are load-dependent statistics related to the number of items in working memory?

Response: Yes, the reviewer is correct as that is indeed what is implied here. We believe that with more items to manipulate in working memory, the network becomes more rigid, i.e. the states are more fixed, thus reducing controllability, It should be noted that controllability is not directly related to the magnitude of coupling but to the structure of the connectivity matrix. We have now clarified these points in the Discussion (page 25).

Unfortunately, the block design of the n-back task precludes examination of load-dependent measures in relation to the number of items in working memory, as there are only two levels of memory load, i.e. 2-back and 0-back.

1.10 In summary, this article deals with an important topic and touches a methodological frontier of applying control theory to task-fMRI inferred connectivity. This unprecedented analysis, however, requires more scrutiny and solidity. I would consider supporting publication if the authors are able to improve the manuscript in this regard in a revised version.

Response: We thank the reviewer for the positive feedback. We have carefully addressed in detail all the issues raised by the reviewers. We hope the revised manuscript meets the high standard of the publication in *Nature Communications*.

Reviewer #2 (Remarks to the Author):

Summary:

In this work, the authors applied causal interaction and network control theory to investigate the brain imaging during working memory tasks in the HCP dataset. They characterized the causal outflow and inflow hubs and examined the dynamic circuit mechanism with control theory. They found the dissociable roles of the SN and FPN in systems control and the asymmetry of control circuits operation during working memory. Overall this is an interesting work with novel perspective of investigating working memory circuits with control methods. The reasoning procedure can potentially be extended to understand other executive functions as well. But it still has many issues to address. I list my comments below.

Motivation & Introduction:

2.1 The current work wants to incorporate three concepts: causal inference, big data reproducibility, and brain control theory. However, from my understanding, these three parts are not coherently integrated. The authors need to better organize the material. My suggestion is that they first apply the causal

inference to identify the hubs of information flow and areas encoding the individual difference; next, they utilize the network control theory to provide a mechanical explanation why the identified nodes play their roles in the circuits from the control perspective rather than simply report their differences in controllability measurement; finally, they could validate their findings based on the replicability.

Response: We thank the reviewer for this suggestion. Based on the reviewer suggestion, we have now organized the Introduction (pages 3-9) and Discussion (pages 18-27) sections of the manuscript along three major concepts: *causal inference, brain control theory and big data reproducibility*. In the sections related causal inference we discuss in order (i) network level causal patterns and differentiation between working-memory task conditions, (ii) node level causal outflow and hubs, (iii) behavioral relevance of causal interaction patterns. Because we examine reproducibility of each major result we would prefer to have report stability results alongside in the key findings so the reader does not have to switch back and forth.

2.2 I believe that the major contribution of the current work is the identification of the directed neural circuits underlining the working memory. So it seems to be inappropriate to firstly talk about the big-data approach in the second paragraph (line 74-80). In addition, they should introduce and discuss the limitation of the undirected network approach here since the working memory task has been investigated for a long time with traditional network approaches.

Response: We thank the reviewer for this suggestion and have now made the suggested changes (pages 3-9). Indeed, the central goal of our work is the identification of the directed neural circuits underlining the working memory using fast-temporal resolution fMRI data.

Results & Discussion:

2.3 To prove that the identified hubs are special for working memory tasks, they should at least include the results for resting state as well for exclusive purpose.

Response: We have now extended our report by including analysis of fixation blocks in the HCP n-back task. Specifically, we examined whether network properties, including undirected weights, directed weights and controllability, are different between task and rest/fixation conditions. First, we examined undirected weights, which is averaging each ROI's inflow and outflow degrees. Paired t-tests showed that all the ROIs and all the networks, SN, FPN and DMN, have significantly smaller undirected weights during rest/fixation than task (2-back and 0-back) conditions ($p < 0.05$, FDR corrected, **Figure 16**).

A. Node-level undirected weights

B. Network-level undirected weights

Figure 16. Undirected weights in 2-back, 0-back and fixation conditions. **(A)** All the ROIs have significantly smaller undirected weights during rest/fixation than task (2-back and 0-back) conditions ($p < 0.05$, FDR corrected). **(B)** SN, FPN and DMN have significantly smaller undirected weights during rest/fixation than task (2-back and 0-back) conditions ($p < 0.05$, FDR corrected).

Next, we examined outflow and inflow weights separately for each condition. We found that IMFG, IFEF, rFEF, IIPL, rIPL and VMPFC had significantly lower outflow weights in rest/fixation than 2-back conditions and rAI, rFEF, rIPL, VMPFC and PCC had significantly lower weights in the rest/fixation than 0-back (all $p < 0.05$, FDR corrected). The IAI, DMPFC, IMFG, IFEF, IIPL, PCC and VMPFC had significantly lower inflow weights in rest/fixation than 2-back and 0-back conditions (all $p < 0.05$, FDR corrected, **Figure 17A**). After merging ROIs into networks, the FPN and DMN had significantly lower outflow weights in rest/fixation than 2-back than 0-back (all $p < 0.05$, FDR corrected). All three network, SN FPN and DMN had significantly lower inflow weights in rest/fixation than 2-back and 0-back conditions (all $p < 0.05$, FDR corrected, **Figure 17B**).

A. Node-level inflow and outflow degree

B. Network-level inflow and outflow degree

Figure 17 Inflow and outflow weights in 2-back, 0-back and fixation conditions. **(A)** IMFG, IFEF, rFEF, IIPL, rIPL and VMPFC had significantly lower outflow weights in rest/fixation than 2-back conditions and rAI, rFEF, rIPL, VMPFC and PCC had significantly lower weights in the rest/fixation than 0-back (all $p < 0.05$, FDR corrected). The lAI, DMPFC, IMFG, IFEF, IIPL, PCC and VMPFC had significantly lower inflow weights in rest/fixation than 2-back and 0-back conditions (all $p < 0.05$, FDR corrected). **(B)** The FPN and DMN had significantly lower outflow weights in rest/fixation than 2-back and 0-back (all $p < 0.05$, FDR corrected). All three network, SN FPN and DMN had significantly lower inflow weights in rest/fixation than 2-back and 0-back conditions (all $p < 0.05$, FDR corrected).

Last, controllability analysis revealed that the strength of controllability during 2-back task are significantly greater than that during 0-back task and fixation ($p < 0.05$, FDR corrected, **Figure 18**) whereas the strength of controllability was not different between 0-back and fixation conditions.

A. Node-level controllability

B. Network-level controllability

Figure 18. Controllability in 2-back, 0-back and fixation conditions. (A) All the ROIs showed significantly lower controllability in the 2-back than 0-back and rest/fixation conditions ($p < 0.05$, FDR corrected). (B) SN, FPN and DMN showed significantly lower controllability in the 2-back than 0-back and res/fixation conditions ($p < 0.05$, FDR corrected).

In summary, these results show that network properties are significantly different between 2-back and rest/fixation conditions: (1) all the ROIs and all the networks, SN, FPN and DMN, have significantly smaller undirected weights during rest/fixation than task (2-back and 0-back) conditions ($p < 0.05$, FDR corrected). (2) All three networks, SN FPN and DMN, have significantly lower inflow weights in rest/fixation than 2-back and 0-back conditions (all $p < 0.05$, FDR corrected). (3) All three networks have significantly smaller controllability during 2-back task are significantly greater than that during 0-back task and fixation ($p < 0.05$, FDR corrected)

2.4 Need an explanation for why DMPFC and the PCC, rather than the AI and MFG are load-dependent.

Response: We found that the net causal influence from AI and MFG are not modulated by working memory load, but the net causal influence from DMPFC and PCC are different between 2-back and 0-back task conditions. AI has been implicated in salience processing, whose activation is among the strongest during cognitive control triggered by external oddball signals (Chen T et al. 2015; Cai WD et al.

2016). In the n-back task, the 2-back and 0-back tasks are not different in external stimuli but the necessity of holding and updating internal representation of stimuli. That may explain why net causal influence of AI is not different between the two conditions. Based on literature in regional activation, we did expect that MFG should have different causal influence to other regions between 2-back and 0-back condition. However, we did not find load-dependency on net causal influence of the MFG. This null effect does not mean that causal interactions between MFG and other regions in the core cognitive control networks are not modulated by working memory load. Indeed, we found the strength of causal interaction between MFG and multiple regions in SN and FPN are different between 2-back and 0-back conditions. For example, the causal interaction between the lMFG and DMPFC, lIPL and rIPL are significantly different between two task conditions and the causal interactions between the rMFG and lAI, rAI, DMPFC, lFEF and rFEF are significantly different between two task conditions. These findings speak to load-dependent causal interactions between MFG and other SN and FPN regions. DMPFC is the key region in the cognitive control system, which has been implicated in a broad range of cognitive control tasks. Lesions and temporal disruption of the DMPFC have shown negative effect on individuals' cognitive control functions. In particular, it has been suggested that DMPFC and the preSMA with which it overlaps, plays an important role in high order of task control, such as switching and updating, which are the major differences in the cognitive processes between 2-back and 0-back task. In the case of the PCC, the differentiation reflects the relatively higher outflow during the low-load working memory condition and is consistent with previous studies demonstrating that dynamic interaction between the PCC and SN and FPN nodes are modulated by the load of attention and cognitive control (Wen XT et al. 2013). Our finding here suggests that net causal influence of the DMN may be related to facilitation of accessing working memory resources.

2.5 Why do they group the nodes into systems to compare the controllability? Is there any relative difference across systems to differentiate with loads? For example, they could normalize the As for 0-back and 2-back scans to eliminate the effect of mean connection strength and prove that the findings were caused by network reconfiguration rather than increased/decreased connection caused by the variation of load.

Response: Grouping nodes into systems can provide us an insight into controllability of specific networks in the larger brain system. MDSI simultaneously estimates the causality between regions (A matrices) under each condition within the same modeling framework. Therefore, the average controllability estimated under the different conditions can be directly compared without the need for additional normalization.

To further address the reviewer's concerns, we repeated the analysis with normalized As and found similar results. Network controllability was evaluated for each node and task condition and entered into an ANOVA with factors working memory load and node. We found a significant main effect of node ($F_{10,736} = 26.73, p < 2.0e-16$) and load ($F_{1,736} = 316.5, p < 2.0e-16$) (**Figure 19A**). Network controllability was lower in the 2-back, compared to the 0-back condition, and this finding held for all nodes (all $ps < 0.001$).

To further evaluate the differential controllability of the three brain networks, we grouped ROIs' controllability scores by their networks and conducted an ANOVA with factors working memory load and network (SN, FPN and DMN). We found a significant main effect of network ($F_{2,736} = 121.8, p < 2.0e-16$), and load ($F_{1,736} = 309.4, p < 2.0e-16$), and a significant interaction between load and network ($F_{2,736} = 28.65, p < 6.24e-13$). SN nodes (rAI, lAI, and DMPFC) had the highest level of controllability in both the 0-back and 2-back tasks. Network controllability was higher in the 0-back, compared to the 2-back condition, and this finding held for all three networks (all $ps < 0.001$, Bonferroni corrected) (**Figure 19B**). Further analysis revealed that the load x network interaction arose from load differences characterized by higher controllability of the SN compared to the FPN ($t = 4.64, p = 4.05e-06$) and the

DMN ($t = 6.46$, $p = 1.82e-10$), and higher controllability of the FPN compared to DMN ($t = 3.55$, $p = 0.0003$).

We then examined the stability of these findings, focusing first on working-memory load dependent differences in network controllability. Higher network controllability on the 0-back, compared to the 2-back, condition achieved a high level of stability (>80%), with sample sizes of $N=30$ or more (**Figure 19C**). Results showing network differences, with SN having the highest network controllability, also showed a high level of stability (>80%) with samples of $N=100$ or more for the 2-back task condition and $N=50$ or more for the 0-back condition.

A. Node-level controllability

B. Network-level controllability

C. Stability of controllability

Figure 19. Controllability analysis based on normalized MDSI values. **(A)** Functional network controllability, assessed in each brain node, was significantly lower in the 2-back, compared to the 0-back, working memory task condition ($p < 0.001$). AI and DMPFC nodes in the SN have significantly higher controllability than FPN and DMN nodes, except for IFEF and IMFG nodes of the FPN in the 2-back task condition and IFEF in the 0-back task condition ($p < 0.001$). **(B)** Functional network controllability, assessed across SN, FPN and DMN nodes, was significantly lower in the 2-back,

compared to the 0-back, working memory task condition ($p < 0.001$). The SN shows significantly higher controllability than the FPN and DMN ($p < 0.001$). (C) Stability analyses revealed stable load effect (0-back > 2-back) and network difference (SN > FPN and SN > DMN). X-axis shows sample size, ranging from 20 to 600. Y-axis shows stability, computed as the probability that the load effect of controllability is significantly different between 2-back and 0-back working memory conditions, and the probability that the SN shows greater controllability than the FPN and DMN in both 2-back and 0-back working memory conditions, in random subsamples drawn from $N=737$ participants.

In summary, we demonstrated that normalization of directed causal weights in the 0-back and 2-back conditions does not change the main findings. These results are now described in page 17.

2.6 Why do they use CCA to identify the dynamic causal interaction patterns? If the response is only 1-dimension or consists of simply accuracy and speed, I think linear regression with/without kernel was enough? (line 330-341).

Response: The advantage of the CCA is that it can effectively capture latent multivariate relationships between brain and behavioral measures ¹⁶, whereas multiple linear regression can only tell us whether each brain feature is related with behavioral measure and such a relationship could be weak. Furthermore, CCA does not require correction for multiple comparisons.

2.7. They claimed that “We suggest that higher causal inflow into the MFG may facilitate maintenance and manipulation of the contents of working memory.” If this is true, there should be observable differences by comparing loads both with and without normalization of connection matrices.

Response: We thank the reviewer for pointing out this apparent inconsistency. We found that the MFG has greater causal inflow during both the 2-back and 0-back conditions. However, between condition differences were not significant, suggesting that greater inflow reflects a more general signaling mechanism independent of task modulation of MFG response. These results were observed with both with and without normalization of connection matrices. We have modified this statement to make it more aligned with our findings.

2.8 The explanation in line 464-466 for PCC’s role in working memory is interesting. It would be better if this could be supported by related results in the context or additional statistical association or causal inferences.

Response: We thank the reviewer for this suggestion. We found increased working memory load is associated with deactivation in the PCC and increased net inflow degree in the PCC. However, we did not find a significant correlation between load-modulated activation change and net inflow degree change in the PCC.

2.9 What additional insight can we obtain when applying predictive models rather than comparing connection-wisely with FDR-FWER correction?

Response: Connection-wise analysis has the advantage that it uncovers the link-specific relationship with behavior. However, connection-wise analysis cannot reveal multivariate patterns associated with behavior. The contributions of each individual links might be weak, especially with FDR-correction.

2.10 For the discussion in line 498-500, it would be better if they add additional explanation what types of reconfiguration cause the decrease of controllability for high load memory tasks.

Response: We observed greater dynamic causal interactions between regions in the SN, FPN and DMN in the 2-back than 0-back working memory task conditions. This indicates extensive dynamic communication within the brain network underlying increased cognitive processing, including sensory encoding, memory maintenance and updating. In such high load context (2-back), controllability of the brain network is reduced compared to a relative low load context (0-back). The decreased controllability is likely associated with dynamic reconfiguration in order to meet fast-changing cognitive processes. For example, an internal representation of stimuli is behaviorally salient in time point t , becomes a confound in time point $t+1$, turns to be important again at time point $t+2$, but not important any more in time point $t+3$. However, we cannot identify specific type of reconfiguration due to the complicate processes in the 2-back working memory task. Like commented by the reviewer #3, the n-back task is not the ideal paradigm to dissociated cognitive sub-processes in working memory.

2.11 For the discussion in 505-507, I am confused by the word “adequate” here. Since the effective control nodes are privileged to be hubs, is there any reason why SN stands out than FPN and DMN based on their nodal roles? Also, this should be supported by group comparison as well.

Response: It has been hypothesized that SN play a crucial role in switching its interaction with other brain networks, in particular FPN and DMN¹⁷. Consistent with this theory, several studies have found that, the anterior insula, the key node in the SN, have significant causal influence other nodes in the FPN and DMN (Sridharan D et al. 2008; Supekar K and V Menon 2012; Chen T *et al.* 2015; Cai WD *et al.* 2016). Here we found that the controllability of SN is significantly greater than that of FPN and DMN in both 0-back and 2-back task conditions, suggesting that SN has more effective control nodes than FPN and DMN. This was supported by group comparison between networks and between nodes in our analysis.

2.12 Figure 1A would be better if the node size was tuned proportionally to the nodal degree.

Response: Figure 1A shows the ROIs used in the current study, which is not related to the nodal degree. **Figure 20** illustrates ROIs whose size are proportionally tuned to their nodal degree, which we have now included in the Supplementary Materials.

Figure 20. Salience Network (SN), Frontal-Parietal Network (FPN) and Default Mode network (DMN) ROIs: 1, left anterior insula (IAI); 2, right anterior insula (rAI); 3, dorsomedial prefrontal cortex (DMPFC); 4, left middle frontal gyrus (lMFG); 5, right middle frontal gyrus (rMFG); 6, left frontal eye field (lFEF); 7, right frontal eye field (rFEF); 8, left intraparietal lobule (lIPL); 9, right intraparietal lobule (rIPL); 10, posterior cingulate cortex (PCC) and 11, ventromedial prefrontal cortex (VMPFC). ROIs’ sizes are proportionally to their node degrees in (A) 2-back and (B) 0-back task conditions.

2.13 It would be great to have Figure 2B visualized on the brain as well.

Response: In response, we have now generated **Figure 21** to illustrate Figure 2B on the brain. However, the 3D brain view of directed connectivity is harder to read than the 2D matrix view because of the large number of connections. We have now included **Figure 21** in the Supplementary Materials.

Figure 21. Dynamic causal interactions between ROIs in Salience Network (SN), Frontal-Parietal Network (FPN) and Default Mode network (DMN): 1, left anterior insula (lAI); 2, right anterior insula (rAI); 3, dorsomedial prefrontal cortex (DMPFC); 4, left middle frontal gyrus (lMFG); 5, right middle frontal gyrus (rMFG); 6, left frontal eye field (lFEF); 7, right frontal eye field (rFEF); 8, left intraparietal lobule (lIPL); 9, right intraparietal lobule (rIPL); 10, posterior cingulate cortex (PCC) and 11, ventromedial prefrontal cortex (VMPFC) in (A) 2-back and (B) 0-back task conditions.

2.14 While the researchers are free to choose their ROIs accordingly, it would be good to use the full-brain model when they talk about the dynamics since the missing regions would also participate the signal progression unless they provide enough evidence that the regions not in the ROIs did not involve in the related information progression.

Response: We thank the reviewer for this comment. Unlike functional and effective connectivity, Bayesnet or GCA, MDSI (like DCM) estimates condition specific causal interactions in the latent neuronal space rather than the observed fMRI. This is done explicitly to minimize the influences of regional variation in HRF which can confound dynamic casual analysis of fMRI data. Currently, MDSI (and related techniques such as DCM) cannot be scaled up to include areas spanning the entire human brain. This tradeoff is necessary to estimate physiologically meaningful causal interactions. The particular ROIs chosen in this study are based on our systems-neuroscience based theoretical models involving key nodes of the SN, CEN and DMN. We have acknowledged the limitation in the study.

2.15 For the classification analysis (line 737-745), I don't think it is necessary to include the GLMNET results here. If so, they may want to compare the weight in the two models as well. In addition, a better setting from my understanding is to have 1/3 of data as test data and perform all experiments on 2/3 of the data. Then they could examine their results and models on the never-seen test data.

Response: We thank the reviewer for this suggestion. However, we cannot directly compare weights from the SVM and GLMNET because they are two different algorithms. SVM identifies a hyperplane in an N-dimension space (n is the number of features, in our application, n=110) that allows differentiation between two groups of data points. GLMNET fits a generalized linear model with regularization for the lasso and elasticnet penalty and provides a solution with sparse parameters (the number of non-zero weights are less than the number of features).

Our analysis used a 10-fold cross validation, which splits data into 10 folds – 9 of 10 folds of data were used as training data and the remaining 1 fold was used as test data. Classification models were trained on the training data and tested on the test data. This procedure was repeated 10 times with each data fold used exactly once as a test set. To avoid data split bias, we repeated the 10-fold cross validation procedure 100 times with random data split into training and test fold each time and reported the mean and standard deviations of cross validation accuracies. The significance of classification accuracy was evaluated using permutations (100 times). The reviewer recommended to use 1/3 of data as training data and perform test on 2/3 of data, which is basically a one-fold of a 3-fold cross validation. Following the reviewer's suggestion, we conducted 3-fold cross validation and replicated the classification results using both SVM (CV Accuracy = 73.3±1%, $p < 0.01$) and GLMNET (CV Accuracy = 73.2±1%, $p < 0.01$).

2.16 I am not sure whether CCA here is necessary. (line 754-764).

Response: CCA¹⁶ allows us to uncover relationships between multivariate features and has been shown to be useful for characterizing latent brain-behavior associations^{18,19}.

2.17 For the stability analysis for multivariate dynamic causal interaction patterns (line 770-784), I feel like in step(6) should be modified by reporting the distribution of correlation between each thresholded dynamic causal interaction matrix and the matrix from the left part rather than the pair of average matrix with original full data as the two parts were overlapped thus not fully proper for replicability test.

Response: We thank the reviewer's suggestion. The stability test is different from the reproducibility test and they have different goals. For the stability test, we asked the question how large sample size is needed to get stable results of the analysis. To achieve this goal, we sub-sampled the full data with varying sub-sample size and computed similarity of the multivariate pattern between sub-samples and multivariate pattern of full sample. If we computed the similarity of multivariate pattern between sub-samples and left-out samples, we might get high similarity at each sub-sample size, but it does not guarantee that multivariate patterns between different sample sizes are similar. In addition, we had conducted reproducibility analysis using cross-validation approaches.

2.18 For the stability analysis for the outflow hubs of the rAI, the inflow hub of the rMFG, the load effect of network controllability and high controllability in SN (line 785-800), it seemed to be auxiliary with the statistical test values. Also, I would prefer to divide the dataset into three parts or train vs. test as above to prove the replicability.

Response: We thank the reviewer's comments. Based on the reviewer's suggestion, we conducted additional analysis splitting data into three subsets and replicated the outflow hubs of the rAI (all $ps < 0.05$, FDR corrected), the inflow hub of the rMFG (all $ps < 0.05$, FDR corrected), the load effect network controllability (all $ps < 0.001$) and high controllability in SN in each subset (all $ps < 0.001$) (**Figures 22, 23, 24**). These results are described in page 17 and in details in Supplementary Materials.

A. subset 1 (1/3 samples)

B. subset 2 (1/3 samples)

C. subset 3 (1/3 samples)

Figure 22. Replication of the outflow hub of the rAI and the inflow hub of the rMFG among the ROIs of the SN, FPN and DMN in both 2-back and 0-back task conditions from three subsets.

A. subset 1 (1/3 samples)

B. subset 2 (1/3 samples)

C. subset 3 (1/3 samples)

Figure 23. Replication of load effect of the controllability in three subsets.

A. subset 1 (1/3 samples)

B. subset 2 (1/3 samples)

C. subset 3 (1/3 samples)

Figure 24. Replication of the highest controllability of SN in three subsets.

Reviewer #3 (Remarks to the Author):

The present study reports network correlates of the visual n-back task in the Human Connectome Project (HCP) dataset, and their relationship to performance. The authors calculated effective connectivity among nodes in the salience (SN), frontoparietal (FPN), and default-mode (DMN) networks. The authors focused on two main metrics: asymmetry of directed influences and network controllability. The authors found a

tendency for SN nodes to project outward with the anterior insula (AI) doing so in a static fashion, and the dorsomedial prefrontal cortex (DMPFC) doing so in a dynamic fashion, increasing its outward influence when cognitive demands increased. The FPN showed a more varied picture with what appears to be more static and dynamic outflow in the left hemisphere, and more static and dynamic inflow in the right hemisphere (*interpretation is my own here*). The various directed interactions were found to be related to performance in 2-back, but not 0-back. The authors also observed that the SN showed higher levels of network controllability, which diminished under high demands. Finally, the authors performed several robustness checks indicating sample sizes needed for stable estimates.

Overall, this is a beautiful set of analyses with some intriguing results. I appreciated not only the importance of what the authors are examining, but also the rigor with which they examined it. However, I was ultimately left wondering “what does it all mean?” As evidence, I typically finish my summary paragraphs (above) with a conclusion, but found myself unable to do so here. Clearly, there are some robust properties of the brain that have been estimated, and they are linked to cognitive behavior. But the manuscript seems to suffer from a lack of a strong theoretical framework. This makes it difficult to tie the results together, and determine how future work should build on these results. Thus, the lack of theory will limit the impact of this work. Therefore, I would suggest that the authors spend some time thinking through the theoretical implications of these data, sandwiching good theory around the beautiful results.

Here are some concrete examples of the theoretical shortcomings. In the introduction, the authors state:

3.1 “We hypothesized that causal interactions involving the SN, FPN, and DMN are dynamically modulated by working memory load and that directed network interaction patterns would differ between the high and low-load working memory conditions.”

I am not sure that anyone would hypothesize differently. Does anyone think that the brain is functionally the same during 0-back and 2-back? If some method that measures function found the brain to be the same across the two conditions, wouldn't one conclude that the measured is flawed? So, this does not setup for a very informative conclusion.

But on the other end:

“We hypothesized that the AI node of the SN would emerge as a causal outflow hub during both the high and low working memory load conditions with significant inflow into FPN and DMN nodes.”

As far as I could tell, there was nothing in the reviewed literature that would lead to such a specific hypothesis. The reviewed literature focused on the network level, rather than the nodal level. This makes the reader wonder why the AI in particular? Why not the DMPFC or other parts of the SN? Why would this be a static pattern? Was the possibility that the AI could be an inflow hub not considered (it, after all, has a lot of afferent connections)? Hence, either there is more specific literature to be tied together or this is a rather peculiar case of Hypothesizing After the Results are Known (HARKing).

Then there is this:

“We further predicted that....the AI node of the SN would emerge as the node with high controllability.”

In this case, not only is this another oddly specific claim about one particular node, it also contradicts the study by Gu, Bassett, and colleagues (2015) that (to my knowledge) introduced controllability metrics to the neuroscience literature, which reported unremarkable controllability of the FPN and CO networks. There are notable differences among the studies (e.g. estimating controllability on fMRI vs DTI), but in the least, one would expect that this prior work would factor into the authors' hypotheses.

Collectively, I would encourage the authors to put as much effort into theory as they do with computation. If they do, I think there is strong potential for impact here. At the present, the manuscript reads as more of an archival report that would be suited for a more specialized outlet (e.g. NeuroImage). I personally think that would be a missed opportunity. But to seize the opportunity, the authors will need to be very clear regarding the mechanistic insight and theoretical advance that these data make.

Response: We thank the reviewer for the positive feedback and suggestions. While task-modulation of dynamic causal interaction in the SN, FPN and DMN is to be expected, the specific pattern of *dynamic causal influences* and which specific links in the SN, FPN and DMN are modulated by working memory load has not been known.

The theoretical focus of the present work lies in a triple network model of cognitive control systems which posits a key role for the salience network (SN) in switching brain networks with the resultant engagement of the FPN and disengagement of the DMN during cognitively demanding tasks (Menon V and LQ Uddin 2010; Cai WD *et al.* 2016). Analysis of causal control circuits has the potential to inform how asymmetries in directed influence allow individual networks or specific brain nodes to control others. The specific hypotheses we test in this context were based on a body of prior work suggesting a key role for the AI node of the SN in switching brain networks during tasks such as the stop signal, Flanker, and oddball tasks (Chen T *et al.* 2015; Cai WD *et al.* 2016). However, working memory load in such cognitive control tasks is minimal and how the SN, FPN and DMN interact causally during working memory is poorly understood. We have revised our hypotheses to state that “the AI node of the SN would emerge as a causal outflow hub during working memory load with significant inflow into FPN and DMN nodes.”

We have also factored findings from Gu *et al.* (Gu S *et al.* 2015) into our hypotheses. As noted by the reviewer, the analysis of how one brain region controls another requires knowledge of asymmetric causal influences between brain regions. Furthermore, as we clarify in Response 1.1., structural topology-based approaches are not good proxies for modeling controllability in dynamical systems. Incorporating functional dynamics into the network models are required to more accurately state space dynamics. We predicted that network controllability would depend on working memory load with lower controllability during high load reflecting less flexibility in network interactions. We further test the hypothesis that the AI node of the SN would emerge as the node with high controllability associated with its dominant pattern of casual outflow. We contrast our theoretical model and framing with the alternative hypothesis of Gu *et al.* (Gu S *et al.* 2015) which posits that the DMN has the highest controllability, which was based on DTI-based symmetric structural connectivity measures.

Finally, it should be noted that our findings of higher averaged controllability of the SN and FPN are more consistent with the known role of these regions in cognition and cognitive control. However, in our study, DMN had the lowest averaged controllability compared to the SN and FPN. The primary cause is the underlying brain features used to estimate controllability in brain networks.

We have incorporated these arguments in the revised manuscript (pages 7-8).

3.2 Towards improving the theoretical framework, here are some specific thoughts that occurred to me while reading the manuscript. I want to make clear that it is not my intention that the authors should bend the manuscript to the way I see things – I believe it is inappropriate for reviewers to act as such. Nevertheless, I figured my specific thoughts might offer some food for thought that could facilitate a clearer theoretical framework. The authors should feel free to pick and choose which, if any, of the following they wish to incorporate:

N-back requires working memory, but studying n-back does not mean one is studying working memory.

N-back is a complex task that requires many processes and a strong argument can be made that it is a cognitive control task more so than a working memory task. Consider that most contemporary working memory tasks find a means to isolate processes involved in maintenance/representation, while the focus on n-back is much more heavy on the control processes that act to update and manipulate representations. Hence, I think the authors start off on the wrong foot by describing working memory. Most working memory experts that I have had discussions with agree that there is very little that the HCP n-back data can do to advance our knowledge of working memory.

Response: We appreciate the reviewer's suggestion. We fully agree that the n-back working memory task is not an appropriate paradigm to dissociate sub-processes such as encoding and manipulation of stimuli in working memory and updating and inhibition in the 2-back task. In the revised manuscript, we have now aligned the theoretical framework in terms of cognitive control processes engaged during working memory. Accordingly, we have noted that our goal is "... to investigate causal dynamic circuit mechanisms underlying cognitive control processes consistently implicated in working memory."

3.3 What these data and analyses are well-positioned to do is tell us about control. All of the analyses that the authors utilize are about control. Asymmetries of directed influence are the essence of how one brain region controls another. A critical failing of much work on control is that it fails to measure such directed influences. Most work uses functional connectivity, which is non-directional, or structural connectivity, which not only fails to have direction, but also fails to have any dynamics (works by Cole, Bassett, Bertolero, Gratton, and others come to mind). Therefore, the authors have measures that can offer insights into control that other methods cannot. In my opinion, this is what sets this work apart and I believe it fits more naturally in a control setting rather than a working memory setting.

Response: We thank the reviewer for this feedback and this indeed was the motivation for our study and the approach we took. We have further clarified this in the revised manuscript, and summarize recent work on control in complex systems which require constraints based on the underlying dynamics (please see also Response 1.1). In the Supplementary Materials we have now included detailed results from our simulations demonstrating that control properties of asymmetric networks are indeed quite different from symmetric ones.

3.4 From a control perspective, the SN and FPN are obviously important networks. However, their relative order in a hierarchical sense is unknown. As far back as classic work by Botvinick and colleagues, the SN (DMPFC in particular), has been thought to drive the FPN (DLPFC in particular). To my knowledge, such directed relationships have never been convincingly established. On the other hand, the FPN is literally referred to as the "control network". Interestingly, work on hierarchical control has often ignored the SN in favor of the FPN in isolation (e.g. Koechlin, Badre, D'Esposito, Nee) despite the theoretical importance of the SN. So, having two different methods (asymmetries of directed interactions, and network controllability) that can examine the relative control properties of the SN and FPN can potentially bring to light the importance of the SN in driving control.

Most work on control either treats both hemispheres of the FPN as homogenous or ignores a hemisphere (often ignoring the right...c.f. works by Badre, Nee, D'Esposito, others). However, the frontal and parietal lobes are not symmetrical, in part due to the language network's presence in the left. A striking aspect of this work is that the right and left FPN clustered into distinct communities with very different inflow/outflow properties. The authors don't seem to key in on this point, but it has important implications for future work. As the authors are aware, right-sided frontal parietal areas are most often implicated in control over responses (e.g. stop signal task). In a hierarchical control sense, responses are low in the hierarchy. Then, I wonder whether these data suggest a control hierarchy that goes SN -> left FPN -> right FPN from motivation (SN) to abstract control (left FPN) to concrete control (right FPN). A speculative possibility that likely requires some further thought and scholarship. But it is this sort

advancing of theory that I think would really push the manuscript to the level of making a substantial impact.

Again, I stress that these are my thoughts colored by my own work, but it's not my manuscript and I certainly would not hold the authors to my perspective. Any way that the authors choose, it just needs to be made clear how theory can advance in light of these data.

Response: We thank the reviewer for this suggestion. We have added these important points in the revised manuscript both in the Introduction (pages 7-8) and Discussion (pages 18, 26).

There have been great interests in understanding control mechanism in prefrontal and parietal regions during working memory and cognitive control for decades. Early research has focused on functional differentiation and organization of prefrontal cortex, led to multiple theories, including material-specific hypothesis^{20,21}, process-specific hypothesis^{22,23}, material and process hypothesis²⁴, and non-specific/domain general hypothesis²⁵, but reached no consensus. Later, several interesting work has suggested functional hierarchical system of the prefrontal cortex in processing information^{26,27,28,29}. Nevertheless, dynamic interactions within the functional hierarchy of the prefrontal cortex during cognitive control process has not been well understood. In addition, the hemispheric difference of the FPN in working memory and cognitive control remains to be studied. The right FPN, along with the salience network, is highly engaged in salience-driven cognitive control task, such as stop-signal task and go/no-go task^{30,31}. It has been suggested that the left FPN is more involved in proactive control or in processing more abstract information³², but further investigation is needed. Few studies have examined dynamic interactions between SN, FPN and DMN during working memory, though some has investigated dynamic interactions during cognitive control^{31,33,34,35,36}. It is noteworthy that n-back working memory task and salience-driven cognitive control paradigms are highly distinctive as the former requires more control (e.g. updating and inhibition) of internal (abstract) representation of external stimuli whereas the latter involves on controlling multiple sensorimotor mapping. Findings from the current study will provide novel understanding of dynamic control mechanism beyond cognitive control of motor response.

To address the question of hemispheric differences in FPN, we conducted additional analyses grouping the ROIs belonging to the left and right FPN and conducted an additional controllability analysis. Indeed, as the reviewer correctly anticipated, we found a hierarchy of controllability with SN > LFPN > RFPN. ANOVA revealed a significant interaction between task and network ($F_{6,4416}=10.43$, $p=1.62e-11$), significant main effect of network ($F_{3,2208}=208.1$, $p<2e-16$) and significant main effect of task ($F_{2,1472}=38.36$, $p<2e-16$). In each task condition, controllability was significantly greater in SN than LFPN, in LFPN than RFPN, and in RFPN than DMN (all $ps<0.01$, Bonferroni corrected). These results point to hemispheric differences and a hierarchy of controllability in lateral frontoparietal cortex. Whether this reflects a control hierarchy from attention capture/motivation (SN) to abstract control (left FPN) to concrete control (right FPN) is an intriguing hypothesis that warrants further investigation²⁸. We have now described these results on page 17 and mentioned the theoretical implication in page 26 of the Discussion.

Figure 25. Functional network controllability, assessed across SN, LFPN, RFPN and DMN nodes, was significantly lower in the 2-back, compared to the 0-back, working memory task condition ($p < 0.001$). The SN shows significantly higher controllability than LFPN, LFPN higher than RFPN, and RFPN higher than DMN ($p < 0.001$).

3.5 *One final thought: I greatly appreciate the analyses the authors performed demonstrating the robustness of their metrics. However, it may be misleading to suggest that sample size is the primary driver of robustness of network level metrics. The Midnight Scan Club data make it clear that one does not need a large sample sizes to get robust measures. Nee, 2019, Communications Biology also makes the point that perhaps the HCP is not representative given the unusually short task scans. As Nee was cited as among the studies using small samples with “inconsistent and contradictory findings”, the authors might be interested in a recent preprint on bioRxiv on that same “small sample” dataset showing excellent consistency of DCM parameters indicating that sample size is not necessary for such robustness (<https://www.biorxiv.org/content/10.1101/2020.03.30.016394v1>). Big data has its own set of flaws, and I would argue that much of, if not most progress in neuroscience has come from carefully collected small samples (consider e.g. animal neurophys, brain damaged patients). Hence, while I absolutely agree that robustness is vital, I believe it is misleading to suggest that large samples are the best way to get it.*

Response: We agree with this view, and have revised our statements in this regard citing the above literature. We have cited the excellent study by Nee et al., 2019³⁷, noting: “While our findings stress the importance of establishing reliability in analysis of causal circuit dynamics, the question of sample size ultimately depends upon experimental design and sufficient individual-level data.” (page 27).

References Cited

1. Ryali S, Chen TW, Supekar K, Menon V. Estimation of functional connectivity in fMRI data using stability selection-based sparse partial correlation with elastic net penalty. *NeuroImage* **59**, 3852-3861 (2012).
2. Sojoudi S, Doyle J. Study of the Brain Functional Network Using Synthetic Data. *Ann Allerton Conf*, 350-357 (2014).
3. Bishop C. *Pattern Recognition and Machine Learning*. Springer (2006).
4. Haber A, Molnar F, Motter AE. State observation and sensor selection for nonlinear networks. *IEEE Trans Control Netw Syst* **5**, 694-708 (2018).

5. Leitold D, Vathy-Fogarassy A, Abonyi J. Controllability and observability in complex networks - the effect of connection types. *Sci Rep* **7**, 151 (2017).
6. Aguirre LA, Portes LL, Letellier C. Structural, dynamical and symbolic observability: From dynamical systems to networks. *PLoS One* **13**, e0206180 (2018).
7. Hespanha JP. *Linear systems theory*. Princeton Press (2009).
8. Stephan KE, Kamper L, Bozkurt A, Burns GA, Young MP, Kotter R. Advanced database methodology for the Collation of Connectivity data on the Macaque brain (CoCoMac). *Philos Trans R Soc Lond B Biol Sci* **356**, 1159-1186 (2001).
9. Srighakollapu MV, Kalaimani R, Pasumarthy R. Optimizing Average Controllability of Networked Systems. *Ieee Decis Contr P*, 2066-2071 (2019).
10. Hamdan AMA, Nayfeh AH. Measures of Modal Controllability and Observability for 1st-Order and 2nd-Order Linear-Systems. *J Guid Control Dynam* **12**, 421-428 (1989).
11. Gu S, *et al.* Controllability of structural brain networks. *Nat Commun* **6**, 8414 (2015).
12. Summers TH, Cortesi FL, Lygeros J. On Submodularity and Controllability in Complex Dynamical Networks (vol 3, pg 91, 2016). *Ieee T Control Netw* **5**, 1503-1503 (2018).
13. Pasqualetti F, Zampieri S, Bullo F. Controllability Metrics, Limitations and Algorithms for Complex Networks. *Ieee T Control Netw* **1**, 40-52 (2014).
14. Dablander F, Hinne M. Node centrality measures are a poor substitute for causal inference. *Sci Rep* **9**, 6846 (2019).
15. Ryali S, Supekar K, Chen T, Menon V. Multivariate dynamical systems models for estimating causal interactions in fMRI. *NeuroImage* **54**, 807-823 (2011).
16. Hotelling H. Relations between two sets of variates. *Biometrika* **28**, 321-377 (1936).
17. Menon V, Uddin LQ. Saliency, switching, attention and control: a network model of insula function. *Brain Struct Funct* **214**, 655-667 (2010).
18. Smith SM, *et al.* A positive-negative mode of population covariation links brain connectivity, demographics and behavior. *Nat Neurosci* **18**, 1565-1567 (2015).

19. Supekar K, Cai WD, Krishnadas R, Palaniyappan L, Menon V. Dysregulated Brain Dynamics in a Triple-Network Saliency Model of Schizophrenia and Its Relation to Psychosis. *Biol Psychiat* **85**, 60-69 (2019).
20. D'Esposito M, Aguirre GK, Zarahn E, Ballard D, Shin RK, Lease J. Functional MRI studies of spatial and nonspatial working memory. *Cognitive Brain Res* **7**, 1-13 (1998).
21. Goldmanrakis PS. Circuitry of primate prefrontal cortex and regulation of behavior by representational memory. In: *Handbook of physiology: the nervous system* (ed^eds F. P). American Physiological Society (1987).
22. Petrides M. Lateral prefrontal cortex: architectonic and functional organization. *Philos T Roy Soc B* **360**, 781-795 (2005).
23. Smith EE, Jonides J. Storage and executive processes in the frontal lobes. *Science* **283**, 1657-1661 (1999).
24. Johnson MK, Raye CL, Mitchell KJ, Greene EJ, Anderson AW. fMRI evidence for an organization of prefrontal cortex by both type of process and type of information. *Cereb Cortex* **13**, 265-273 (2003).
25. Duncan J, Owen AM. Common regions of the human frontal lobe recruited by diverse cognitive demands. *Trends Neurosci* **23**, 475-483 (2000).
26. Badre D, D'Esposito M. Functional magnetic resonance imaging evidence for a hierarchical organization of the prefrontal cortex. *J Cogn Neurosci* **19**, 2082-2099 (2007).
27. Badre D, Hoffman J, Cooney JW, D'Esposito M. Hierarchical cognitive control deficits following damage to the human frontal lobe. *Nat Neurosci* **12**, 515-522 (2009).
28. Nee DE, D'Esposito M. The hierarchical organization of the lateral prefrontal cortex. *Elife* **5**, (2016).
29. Nee DE, D'Esposito M. Causal evidence for lateral prefrontal cortex dynamics supporting cognitive control. *Elife* **6**, (2017).
30. Cai W, *et al*. Hyperdirect insula-basal-ganglia pathway and adult-like maturity of global brain responses predict inhibitory control in children. *Nat Commun* **10**, 4798 (2019).
31. Cai W, Ryali S, Chen T, Li CS, Menon V. Dissociable roles of right inferior frontal cortex and anterior insula in inhibitory control: evidence from intrinsic and task-related functional parcellation, connectivity, and response profile analyses across multiple datasets. *The Journal of neuroscience : the official journal of the Society for Neuroscience* **34**, 14652-14667 (2014).

32. Kiyonaga A, Korb FM, Lucas J, Soto D, Egner T. Dissociable causal roles for left and right parietal cortex in controlling attentional biases from the contents of working memory. *NeuroImage* **100**, 200-205 (2014).
33. Cai WD, Chen TW, Ide JS, Li CSR, Menon V. Dissociable Fronto-Operculum-Insula Control Signals for Anticipation and Detection of Inhibitory Sensory Cues. *Cereb Cortex* **27**, 4073-4082 (2017).
34. Chen T, Michels L, Supekar K, Kochalka J, Ryali S, Menon V. Role of the anterior insular cortex in integrative causal signaling during multisensory auditory-visual attention. *Eur J Neurosci* **41**, 264-274 (2015).
35. Ham T, Leff A, de Boissezon X, Joffe A, Sharp DJ. Cognitive control and the salience network: an investigation of error processing and effective connectivity. *The Journal of neuroscience : the official journal of the Society for Neuroscience* **33**, 7091-7098 (2013).
36. Jilka SR, *et al.* Damage to the Salience Network and interactions with the Default Mode Network. *The Journal of neuroscience : the official journal of the Society for Neuroscience* **34**, 10798-10807 (2014).
37. Nee DE. fMRI replicability depends upon sufficient individual-level data. *Commun Biol* **2**, (2019).

Reviewer #1 (Remarks to the Author):

The authors are to be commended for their rigorous and in-depth revisions, which comprise several changes to the text and many new analyses to support their conclusions. With one exception (see below), all of my questions and concerns have been thoroughly addressed.

Authors' response to my major point 1.1:

The authors have explained the advantages of MDSI and the practical reason for not conducting a whole-brain level analysis. However, these facts do not exempt MDSI from being prone to the confounding factors that are unobserved. One should be careful when interpreting the result of "causality"/connectivity (Das & Fiete, Nat Neurosci 2020) and controllability.

The difference of presumed connectivity between the 2-back and 0-back/resting state condition could be due to the un-observed confounding common cause nodes being controlled in the 2-back condition, thus removing some otherwise presumed connectivity edges. In this scenario, the true connectivity has not changed, but depending on the confounder node being controlled or not (this is what I meant by a state being fixed here in comment 1.1 or a network being rigid in comment 1.9), the presumed connectivity matrices are different.

One could of course argue that the presumed connectivity, even caused by a confounder, is a good measure of how the network operates. However, this presumed connectivity, let alone causality, is different from the connectivity many neuroscientists would intuitively think of. I therefore suggest (related to comment 1.6) the authors update the text to distinguish intrinsic and modulatory connectivity and to clarify that the difference between the 0-back and 2-back condition is the difference of their modulatory connectivity, which is possibly caused by the un-observed confounder mentioned above.

Providing these changes are implemented, I would be happy to support publication of this manuscript in Nature Communications.

Das, A., Fiete, I.R. Systematic errors in connectivity inferred from activity in strongly recurrent networks. Nat Neurosci 23, 1286–1296 (2020). <https://doi.org/10.1038/s41593-020-0699-2>

Reviewer #2 (Remarks to the Author):

The authors addressed most of my concerns with sufficient work for the reversion. I only have a few questions left.

1. The use of CCA. I believe I understand what CCA is and my concern is that only two variables are included here thus the CCA would not fit properly here. In addition, in terms of the multi-test correction, the CCA does need to be corrected. Considering the case you only have the random variable, due to the distribution of random correlations, the identified maximum of correlation could always be significant.

2. For the comparison of GLMNET and SVM, as the author said that SVM identified the hyperplane and GLMNET fit the linear model, both of them looked for a linear model to separate the samples with some constraints. Thus the two models are not totally independent and the weight can be interpreted similarly.

Reviewer #3 (Remarks to the Author):

The authors have been highly responsive to my previous concerns. The necessary theoretical framing has been given upfront, and the intricate analyses that follow can now be much better appreciated. I applaud the efforts of the authors in this regard. In my reading, it seemed as though the authors "lost steam" a bit towards the end. That is, I would have liked to have seen the

authors better connect the description of their results to theory in the Discussion to fully tie things together. They have put substantial work into this manuscript, and if the reader is left wondering what they have learned, it would truly be a missed opportunity. My remaining comments are all of the minor variety, and I hope that they help the authors complete this excellent manuscript:

p. 20 lines 451-459, the authors describe directed influences that distinguish 2-back from 0-back and conclude "These results demonstrate how working memory alters causal dynamics in the SN-FPN-DMN cognitive control system." I did not feel as though this was demonstrated. Although the connection between the measurements we make and mental process can be somewhat tenuous, some connection must be made so that understanding can occur on the mental process level. In this case, "working memory" is a mental process and "causal dynamics" are measurements. To understand something about working memory or cognitive control, the authors will need to infer something about what greater influences during 2-back relative to 0-back mean on a mental process level. They set this up in the introduction suggesting a framework wherein the SN switches brain networks to engage the FPN and disengage the DMN. The described results appear consistent with that framework. Now, the framework needs to be explicitly tied to n-back and the demands therein so that the reader understands what it means for our understanding of mental function.

p. 22 lines 501-502: "We suggest that higher causal inflow into the MFG may facilitate maintenance and manipulation of the contents of working memory." Elaborating how (at a descriptive level) causal inflow facilitates maintenance and manipulation would help give this sentence some teeth.

P 24 lines 544-545: "Reduced outflow signals are a likely mechanism by which the DMN facilitates access to working memory resources thereby enhancing task performance." Although I can infer elaborative accounts of the two examples above, here I am admittedly baffled. Access, in a colloquial sense, is typically granted by removal of a barrier (e.g. opening a door). So, do the authors mean to say that the DMN grants access by removing its influence? If so, the phrasing is a bit odd, because we typically consider that an entity that grants access is something external to the barrier itself. Some elaboration would be welcome.

Reviewer #1 (Remarks to the Author):

The authors are to be commended for their rigorous and in-depth revisions, which comprise several changes to the text and many new analyses to support their conclusions. With one exception (see below), all of my questions and concerns have been thoroughly addressed.

Authors' response to my major point 1.1:

The authors have explained the advantages of MDSI and the practical reason for not conducting a whole-brain level analysis. However, these facts do not exempt MDSI from being prone to the confounding factors that are unobserved. One should be careful when interpreting the result of “causality”/connectivity (Das & Fiete, Nat Neurosci 2020) and controllability.

The difference of presumed connectivity between the 2-back and 0-back/resting state condition could be due to the un-observed confounding common cause nodes being controlled in the 2-back condition, thus removing some otherwise presumed connectivity edges. In this scenario, the true connectivity has not changed, but depending on the confounder node being controlled or not (this is what I meant by a state being fixed here in comment 1.1 or a network being rigid in comment 1.9), the presumed connectivity matrices are different.

One could of course argue that the presumed connectivity, even caused by a confounder, is a good measure of how the network operates. However, this presumed connectivity, let alone causality, is different from the connectivity many neuroscientists would intuitively think of. I therefore suggest (related to comment 1.6) the authors update the text to distinguish intrinsic and modulatory connectivity and to clarify that the difference between the 0-back and 2-back condition is the difference of their modulatory connectivity, which is possibly caused by the un-observed confounder mentioned above.

Providing these changes are implemented, I would be happy to support publication of this manuscript in Nature Communications.

Das, A., Fiete, I.R. Systematic errors in connectivity inferred from activity in strongly recurrent networks. Nat Neurosci 23, 1286–1296 (2020). <https://doi.org/10.1038/s41593-020-0699-2>

Response: We thank the reviewer for this suggestion and the reference. We agree with the sentiments expressed. As suggested, we have updated the text to clarify that the difference between the 0-back and 2-back condition is the difference of their modulatory connectivity and that context-dependent causal interactions could be influenced by unobserved confounds (page 20).

Reviewer #2 (Remarks to the Author):

The authors addressed most of my concerns with sufficient work for the reversion. I only have a few questions left.

1. The use of CCA. I believe I understand what CCA is and my concern is that only two variables are included here thus the CCA would not fit properly here. In addition, in terms of the multi-test correction, the CCA does need to be corrected. Considering the case you only have the random variable, due to the distribution of random correlations, the identified maximum of correlation could always be significant.

Response: We thank the reviewer for this comment. We used the CCA to investigate relationships between multivariate brain features and behavioral variables, including 110 variables on the brain measure (i.e. MDSI weights) and 2 variables on the behavioral measure (Accuracy and RT). The validity of the CCA model is independent of the number of variables included in the model. We conducted CCA on 0-back and 2-back conditions separately, and found that our reported results held after multiple comparison correction. We have clarified this in the revised manuscript (Page 14). Note that we found significant canonical correlation between brain and behavior measures in 2-back but not in 0-back conditions, suggesting that that that results do not arise from random correlations. Moreover, rigorous permutation testing confirmed our results.

2. For the comparison of GLMNET and SVM, as the author said that SVM identified the hyperplane and GLMNET fit the linear model, both of them looked for a linear model to separate the samples with some constraints. Thus the two models are not totally independent and the weight can be interpreted similarly.

Response: We agree with the reviewer that both GLMNET and SVM with linear kernel are linear models for supervised learning. Because GLMNET fits a generalized linear model with regularization for the lasso and elasticnet penalty and provides a solution with sparse parameters, the number of non-zero weights in the GLMNET model are much less than the number of features in the SVM. We had originally provided both results to demonstrate robustness of our findings. In order to address the reviewer's concern about redundancy, we have now included only the SVM model results in the revised manuscript.

Reviewer #3 (Remarks to the Author):

The authors have been highly responsive to my previous concerns. The necessary theoretical framing has been given upfront, and the intricate analyses that follow can now be much better appreciated. I applaud the efforts of the authors in this regard. In my reading, it seemed as though the authors "lost steam" a bit towards the end. That is, I would have liked to have seen the authors better connect the description of their results to theory in the Discussion to fully tie things together. They have put substantial work into this manuscript, and if the reader is left wondering what they have learned, it would truly be a missed opportunity. My remaining comments are all of the minor variety, and I hope that they help the authors complete this excellent manuscript:

p. 20 lines 451-459, the authors describe directed influences that distinguish 2-back from 0-back and conclude "These results demonstrate how working memory alters causal dynamics in the SN-FPN-DMN cognitive control system." I did not feel as though this was demonstrated. Although the connection between the measurements we make and mental process can be somewhat tenuous, some connection must be made so that understanding can occur on the mental process level. In this case, "working memory" is a mental process and "causal dynamics" are measurements. To understand something about working memory or cognitive control, the authors will need to infer something about what greater influences during 2-back relative to 0-back mean on a mental process level. They set this up in the introduction suggesting a framework wherein the SN switches brain networks to engage the FPN and disengage the DMN. The described results appear consistent with that framework. Now, the framework needs to be explicitly tied to n-back and the demands therein so that the reader understands what it means for our understanding of mental function.

Response: We thank the reviewer's comment. We have modified the sentence to "These results demonstrate that modulation of cognitive load in working memory task alters causal dynamics in the SN-FPN-DMN cognitive control system." (Page 20)

p. 22 lines 501-502: "We suggest that higher causal inflow into the MFG may facilitate maintenance and manipulation of the contents of working memory." Elaborating how (at a descriptive level) causal inflow facilitates maintenance and manipulation would help give this sentence some teeth

Response: We thank the reviewer for this comment. We have now elaborated on this: "We suggest that higher causal inflow into the MFG may facilitate working memory in two ways. First, causal signals may activate MFG regions known to play a critical role in maintenance and manipulation of the contents of working memory. Second, higher inflow reflects convergence of signals associated with external stimuli, internal representations and task rules that are needed for working memory. Further studies with simultaneous electrophysiological recordings are required to test this hypothesis." (Page 22)

P 24 lines 544-545: "Reduced outflow signals are a likely mechanism by which the DMN facilitates access to working memory resources thereby enhancing task performance." Although I can infer elaborative accounts of the two examples above, here I am admittedly baffled. Access, in a colloquial sense, is typically granted by removal of a barrier (e.g. opening a door). So, do the authors mean to say that the DMN grants access by removing its influence? If so, the phrasing is a bit odd, because we typically consider that an entity that grants access is something external to the barrier itself. Some elaboration would be welcome.

Response: We agree with the reviewer that we were not clear here and that this sentence does not accurately describe what we meant. We now have changed it in the revised manuscript to the following: "Our findings demonstrate that deactivation of the PCC is accompanied by decreased causal outflow from the PCC (Supplementary Figure 6). Reduced outflow signals reflect disengagement of the DMN from the FPN and SN which facilitates access to working memory resources thereby enhancing task performance." (Page 24)

Reviewer #1 (Remarks to the Author):

The authors have provided a half-sentence (l. 459/460) in response to my remaining comment on the confounds of the presented results. I would appreciate a full paragraph on this matter with a more serious treatment of the possible un-observed physiological processes that their “dynamic causal interactions” could actually reflect.

Reviewer #2 (Remarks to the Author):

The authors have addressed my concerns and I have no further questions.

Reviewer #3 (Remarks to the Author):

The authors have made light edits in accordance with my previously raised minor concerns about tying in their results more directly with theoretical ideas they introduce at the beginning of the manuscript. Although I think that the authors could have been more expansive in their responses, I do not wish to delay this manuscript from publication further. I hope that the authors do take some time to reflect on their work and deeper theoretical implications when time permits. The utility of cognitive neuroscience will be limited if we content ourselves with being descriptive. Overall, however, my enthusiasm for this manuscript and the rigorous approach of the authors remains high.

Reviewer #1

The authors have provided a half-sentence (l. 459/460) in response to my remaining comment on the confounds of the presented results. I would appreciate a full paragraph on this matter with a more serious treatment of the possible un-observed physiological processes that their “dynamic causal interactions” could actually reflect.

Response: We apologize that for space reasons we had not included a detailed discussion of this issue. We deeply appreciate the reviewer’s suggestion as this is clearly a challenge for all future research, both neuronal (as noted by Das and Fiete) and brain imaging (as noted by Kording). We would also like to point out that MDSI does not falsely estimate causal interactions between the nodes which are common inputs to a node. MDSI uses a vector autoregressive model in the latent space which regresses out the influence of other regions while estimating the causal interactions between two regions ^{1,2,3}. Nevertheless, MDSI and other causal estimation methods do not account for the influences of inputs from the nodes not included in the model on the causal interactions between the nodes in the model. New approaches need to be developed to address this issue. As suggested, we have now included a deeper treatment of this question on pages 28-29. Specifically, we note:

Note on interpreting causality and the effects of unobserved variables

The present study has leveraged advances in state-space models to jointly infer causal interactions between brain regions without the confounding influences of regional variation in hemodynamic response function or the need to test multiple network models ². While this provides distinct advantages over techniques such as Granger causal analysis, dynamic causal modeling and transfer entropy, two issues need to be highlighted. First, at present no (validated) methods exist that can extend causal network analysis, controlling for hemodynamic confounds, to the whole-brain level, spanning 350 or more anatomically distinct regions, due to the sheer number of parameters that need to be estimated. Instead, our approach has focused on probing causal dynamics associated with three key prefrontal cortex networks consistently implicated in cognitive control and working memory. Second, relatedly, this raises the question of whether findings could be influenced by unobserved confounds because erroneous inferences on connectivity (causal or otherwise) can occur when data from a limited set of brain regions or neuronal populations is used in data analysis ⁴, a problem that is further confounded when there is a mismatch between the true network dynamics and the model used for inference ⁵. There currently are no good solutions to these problems, short of extensive invasive manipulations to each processing unit spanning the entire brain. We have addressed this challenge in the present study by, as summarized in the previous section, conducting extensive reproducibility, stability and cross-validation analyses using a large sample size, the only realistic approach with real-world non-invasive human brain imaging data.

Reviewer #3:

The authors have made light edits in accordance with my previously raised minor concerns about tying in their results more directly with theoretical ideas they introduce at the beginning of the manuscript. Although I think that the authors could have been more expansive in their responses, I do not wish to delay this manuscript from publication further. I hope that the authors do take some time to reflect on their work and deeper theoretical implications when time permits. The utility of cognitive neuroscience will be limited if we content ourselves with being descriptive. Overall, however, my enthusiasm for this manuscript and the rigorous approach of the authors remains high.

Response: We appreciate the continued enthusiasm of the reviewer, and apologize that we did not include a detailed discussion due to space limitations. We thank the reviewer for pushing us to clarify further the theoretical implications, which we have now done in the context of the theoretical model of cognitive control systems we sought to probe. As suggested, we have now included a deeper treatment on page 27. Specifically, we note:

Robust mechanisms underlying triple network model of dynamic cognitive control

The present study has focused on a theoretical model of prefrontal cortex networks that support flexible cognitive control functions^{6,7}. Our findings based on asymmetric patterns of directed causal influence between key nodes of the SN, FPN, and DMN as well as their functional controllability are consistent with the triple network model of cognitive control, and shed further light on the underlying mechanisms and their robustness. First, the AI showed the highest causal outflow among all nodes suggesting a dominant role of this key SN node in driving dynamic interactions between regions between the SN, FPN and DMN. Second, high levels of causal inflow into the MFG suggests that this FPN node is crucial for integrating cognitive control signals from multiple other prefrontal and parietal cortical regions. Third, disengagement of the DMN from the SN and FPN under high cognitive load is marked by reduced outflow signals from the PCC. Fourth, the SN demonstrated the highest controllability, indicating a prominent role in driving network interactions. Fifth, these findings were replicable and stable, as discussed below. Taken together, these findings provide new insights into the mechanisms by which prefrontal cortex networks implement cognitive control and facilitate rapid system reorganization to meet cognitive task demands. Further studies are needed to clarify the role of SN, FPN and DMN in the context of models of hierarchical control associated with different the levels of abstraction⁸ and multiple-demand systems engaged by diverse cognitive demands⁹.

Reference

1. Ryali S, *et al.* Multivariate dynamical systems-based estimation of causal brain interactions in fMRI: Group-level validation using benchmark data, neurophysiological models and human connectome project data. *J Neurosci Meth* **268**, 142-153 (2016).
2. Ryali S, *et al.* Combining optogenetic stimulation and fMRI to validate a multivariate dynamical systems model for estimating causal brain interactions. *NeuroImage* **132**, 398-405 (2016).
3. Ryali S, Supekar K, Chen T, Menon V. Multivariate dynamical systems models for estimating causal interactions in fMRI. *NeuroImage* **54**, 807-823 (2011).
4. Mehler DMA, Kording KP. The lure of misleading causal statements in functional connectivity research. *arXiv*, (2018).
5. Das A, Fiete IR. Systematic errors in connectivity inferred from activity in strongly recurrent networks. *Nat Neurosci* **23**, 1286-1296 (2020).
6. Cai WD, Chen TW, Ryali S, Kochalka J, Li CSR, Menon V. Causal Interactions Within a Frontal-Cingulate-Parietal Network During Cognitive Control: Convergent Evidence from a Multisite-Multitask Investigation. *Cereb Cortex* **26**, 2140-2153 (2016).
7. Menon V, Uddin LQ. Saliency, switching, attention and control: a network model of insula function. *Brain Struct Funct* **214**, 655-667 (2010).

8. Badre D, D'Esposito M. Functional magnetic resonance imaging evidence for a hierarchical organization of the prefrontal cortex. *J Cogn Neurosci* **19**, 2082-2099 (2007).
9. Duncan J. The multiple-demand (MD) system of the primate brain: mental programs for intelligent behaviour. *Trends Cogn Sci* **14**, 172-179 (2010).

Reviewer #1 (Remarks to the Author):

The authors have addressed my last remaining point. Congratulations on a very nice paper.

Reviewer #3 (Remarks to the Author):

The authors have satisfactorily addressed my remaining minor concerns